# High-entropy intermetallics on ceria as efficient catalysts for the oxidative dehydrogenation of propane using $CO_2$

Feilong Xing [1], Jiamin Ma[1], Ken-ichi Shimizu [1] & Shinya Furukawa [1,2] ✉

The oxidative dehydrogenation of propane using $CO_2$ ($CO_2$-ODP) is a promising technique for high-yield propylene production and $CO_2$ utilization. The development of a highly efficient catalyst for $CO_2$-ODP is of great interest and benefit to the chemical industry as well as net zero emissions. Here, we report a unique catalyst material and design concept based on high-entropy intermetallics for this challenging chemistry. A senary (PtCoNi)(SnInGa) catalyst supported on $CeO_2$ with a PtSn intermetallic structure exhibits a considerably higher catalytic activity, $C_3H_6$ selectivity, long-term stability, and $CO_2$ utilization efficiency at 600 °C than previously reported. Multi-metallization of the Pt and Sn sites by Co/Ni and In/Ga, respectively, greatly enhances propylene selectivity, $CO_2$ activation ability, thermal stability, and regenerable ability. The results obtained in this study can promote carbon-neutralization of industrial processes for light alkane conversion.

Global demand for propylene, a raw material for petrochemicals, has resulted in a significant supply-demand gap as the feedstock has shifted from naphtha to shale gas production. As a result, increasing the supply of propylene from shale gas is currently highly desired in highly efficient technologies[1–3]. Direct dehydrogenation of propane (DDP) is an appealing on-purpose approach in principle. However, high propylene yields are typically obtained at high reaction temperatures (typically ≥ 600 °C) owing to its endothermicity[4]. Furthermore, under such harsh conditions, undesired side reactions (over-dehydrogenation and C–C cracking) occur, resulting in coke accumulation and, eventually, the catalysts in use must be regenerated by coke combustion in a short period, decreasing productivity for a stable and continuous supply of propylene[2]. The oxidative dehydrogenation of propane using $CO_2$ as a soft oxidant ($CO_2$-ODP: $C_3H_8 + CO_2 \rightarrow C_3H_6 + CO + H_2O$)[5,6] is a promising alternative to DDP, where $CO_2$ can remove carbon via the reverse Boudouard reaction ($CO_2 + C \rightarrow 2CO$)[7]. In comparison to ODP using $O_2$, $CO_2$-ODP avoids oxidation of propylene and the catalyst. Moreover, converting $CO_2$ into a value-added chemical is an appealing approach for a carbon neutral and green sustainable society.

Many researchers have focused in recent decades on the use of metal oxides such as $V_2O_5$[8], $Cr_2O_3$[9,10], $Ga_2O_3$[11], and $In_2O_3$[12] as good $CO_2$-ODP catalysts. Despite the fact that chromium oxide species dispersed on mesoporous silica showed the highest catalytic activity with moderate propylene selectivity (ca. 80%), the $CO_2$ utilization efficiency (excess $CO_2$ is necessary) and catalyst stability were low. Transition metal-based catalysts like Pd, $Fe_3Ni$, and Pt–Co–In[5,13,14] have been receiving more attention in $CO_2$-ODP, owing to their multifunctional properties for simultaneous activation of propane and $CO_2$[15–17]. C–H activation of propane and $CO_2$ reduction can be mediated by noble metals (Pt and Pd)[18,19] and the late 3d transition metals (Co, Ni, and Cu)[20,21], respectively; thus, an appropriate combination of these metals allows for the construction of a dual functional catalyst for propane and $CO_2$ conversion. Using $CeO_2$ or Ce-based oxides as a support of the metallic phase is also a promising way for $CO_2$-ODP owing to various promotional effects such as Mars-van Kreveren-type $CO_2$ activation[13,22,23] or coke combustion[14,24], and strong metal-support interaction to tune the character of the active phase[25,26]. We recently reported that the Pt–Co–In/$CeO_2$ catalyst exhibited remarkably high catalytic activity and $CO_2$ utilization efficiency[14]. However, irreversible

[1]Institute for Catalysis, Hokkaido University, N21, W10, Sapporo 001-0021, Japan. [2]Japan Science and Technology Agency, PRESTO, Chiyodaku, Tokyo 102-0076, Japan. ✉e-mail: furukawa@cat.hokudai.ac.jp

catalyst deactivation still occurred, which was likely caused by the accumulation of incombustible coke and nanoparticle sintering. Therefore, it is highly desired to develop a more efficient catalyst that is stable at high temperatures, very selective to minimize side reactions, and can burn coke more efficiently.

To address this challenge, we developed a novel catalyst design concept based on high-entropy intermetallics (HEIs)[27,28]. HEIs are multi-metallic alloys with five or more elements and specific crystal structures originating from the parent binary intermetallics (Fig. 1). HEIs, as opposed to high-entropy alloys (HEAs) with random atomic distribution, can provide ordered reaction environments suitable for CO$_2$-ODP. To minimize unwanted side reactions, the intermetallic PtSn, which is a highly selective DDP catalyst, was chosen as the parent platform of HEI to minimize undesired side reactions. However, alloying with typical metals is known to significantly decrease the CO$_2$ activation ability of Pt. Therefore, PtSn's Pt site was then partially substituted with Ni and Co to incorporate metals more capable of CO$_2$ activation. This has another merit to further dilute Pt–Pt sites for higher selectivity. On the other hand, the Sn site of PtSn was partially replaced with In and Ga, increasing mixing entropy and the resulting thermodynamic stability. When the formation entropy of the parent intermetallics is significantly negative (PtSn: $\Delta H_f = -74.0 \, \text{kJmol}^{-1}$)[29], site-specific multi-metallization is possible. The higher number of constituent metals also enhances the kinetic stability of nanoparticles owing to the sluggish diffusion effect, which prevents sintering or segregation[30]. Furthermore, CeO$_2$ was used as a catalyst support for facile CO$_2$ capture and coke combustion because it is basic and can release oxygen, which is advantageous for these purposes[5,31]. In this case, we demonstrate that a PtCoNiInGaSn/CeO$_2$ catalyst with a HEI structure (donated as HEI/CeO$_2$) based on intermetallic PtSn can act as a highly efficient CO$_2$-ODP catalyst, exhibiting exceptional catalytic activity, C$_3$H$_6$ selectivity, coke resistance, thermal stability, and CO$_2$ utilization stability.

## Results

### Structure characterization of the catalysts

The HEI/CeO$_2$ catalyst was prepared using a conventional co-impregnation method with CeO$_2$ as the support. Figure 2a shows a high-angle annular dark field scanning transmission electron microscopy (HAADF-STEM) image of HEI/CeO$_2$ as well as the corresponding elemental maps obtained through energy-dispersive X-ray (EDX) analysis. Metallic nanoparticles were not visible in the HAADF-STEM image, most likely due to insufficient Z-contrast between Ce and the

other metals. The elemental maps of Pt, Co, Ni, Sn, In, and Ga, on the other hand, clearly indicated the presence of a small (~4 nm) multi-metallic nanoparticle on the CeO$_2$ support. In the multi-metallic nanoparticles, the transition and typical metals atomic ratios ((Pt + Co + Ni)/(Sn + In + Ga)) were approximately 1:1 (Supplementary Figs. 1–3). The crystal structures of the HEI catalysts were then analyzed using X-ray diffraction (XRD). When Al$_2$O$_3$ or SiO$_2$ was used as a catalyst support for HEI, two intense peaks appeared at 41.8° and 44.1°, which corresponded to the 102 and 110 diffractions of intermetallic PtSn (Fig. 2b). The shifts in diffraction angles from pristine PtSn can be attributed to changes in the lattice constant caused by site-specific multi-metallization. However, the corresponding diffractions were not observed for HEI/CeO$_2$. One possible interpretation is that the reflection of X-ray on the nanoparticles was significantly weakened by the strong X-ray scattering by Ce, of which atomic weight and the scattering factor are much larger than Si and Al. Thus, HAADF-STEM and XRD did not provide convincing information for the CeO$_2$-supported catalyst, as is commonly reported in the literature. As a result, X-ray adsorption fine structure (XAFS) analysis was performed to obtain additional structural information. The Pt L$_{III}$-, Co K-, Ni K-, Sn K-, and In K-edge X-ray absorption near edge spectra (XANES) of HEI/CeO$_2$ matched those of the corresponding reference foils, indicating that these metals are reduced to zero-valent states (Fig. 2c). In contrast, the Ga K-edge XANES of HEI/CeO$_2$ were similar to those of Ga$_2$O$_3$, indicating that Ga species in HEI were mostly oxidized. We also performed H$_2$ temperature-programmed reduction, in which the metal precursors were reduced below 600 °C (Supplementary Fig. 4). Considering the order of reduction potentials (Ga$^{3+}$ + 3$e^-$ → Ga$^{(0)}$: −0.55 V vs. SHE < CeO$_2$ + $e^-$ → Ce$_2$O$_3$: −0.36 V < In$^{3+}$ + 3$e^-$ → In$^{(0)}$: −0.34 V, see Supplementary Table 1 for more information on other metals), it is likely that only oxophilic Ga was reoxidized by CeO$_2$ lattice oxygen, even if it could be reduced by H$_2$. Figure 2d shows the Pt L$_{III}$-edge EXAFS oscillation of Pt foil, PtSn/SiO$_2$, and HEI/CeO$_2$. The oscillation feature of HEI/CeO$_2$ was similar to that of intermetallic PtSn but completely different from that of fcc Pt, indicating that Pt species in HEI are present in a PtSn-like structure. Figure 2e shows the Fourier transform of the In K-edge EXAFS of HEI/CeO$_2$, where three peaks assignable to In–O, In–Co/Ni, and In–Pt were distinctly observed (see Supplementary Figs. 5–7, and Supplementary Table 2 for details of curve-fitting). The observation of In–Pt and In–Co/Ni scatterings suggests that In is also present in the PtSn-like structure (at the Sn site) and that Co(Ni) is doped into the Pt site. The small contribution of oxygen could be attributed to interactions with CeO$_2$'s lattice oxygen. We also

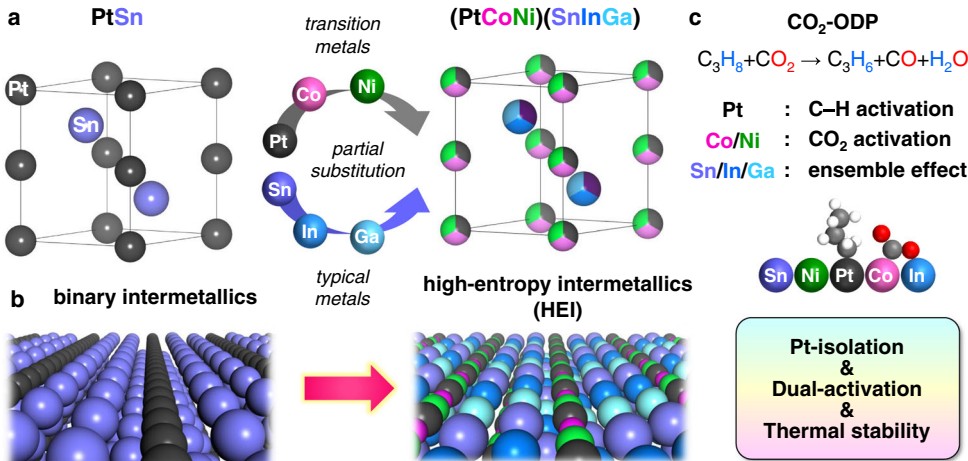

**Fig. 1 | Aim of this work. a** Catalyst design concept based on HEI. Pt and Sn sites in intermetallic PtSn (hexagonal NiAs-type structure, space group: P6$_3$/mmc) are partially substituted by Co/Ni and In/Ga, respectively, forming a PtSn-type HEI (PtCoNi) (SnInGa). **b** Atomic arrangement of the most stable (110) surfaces of PtSn (left) and (PtCoNi)(SnInGa) HEI (right). **c** The role of each metal and the effect of multi-metallization on the catalysis of CO$_2$-ODP.

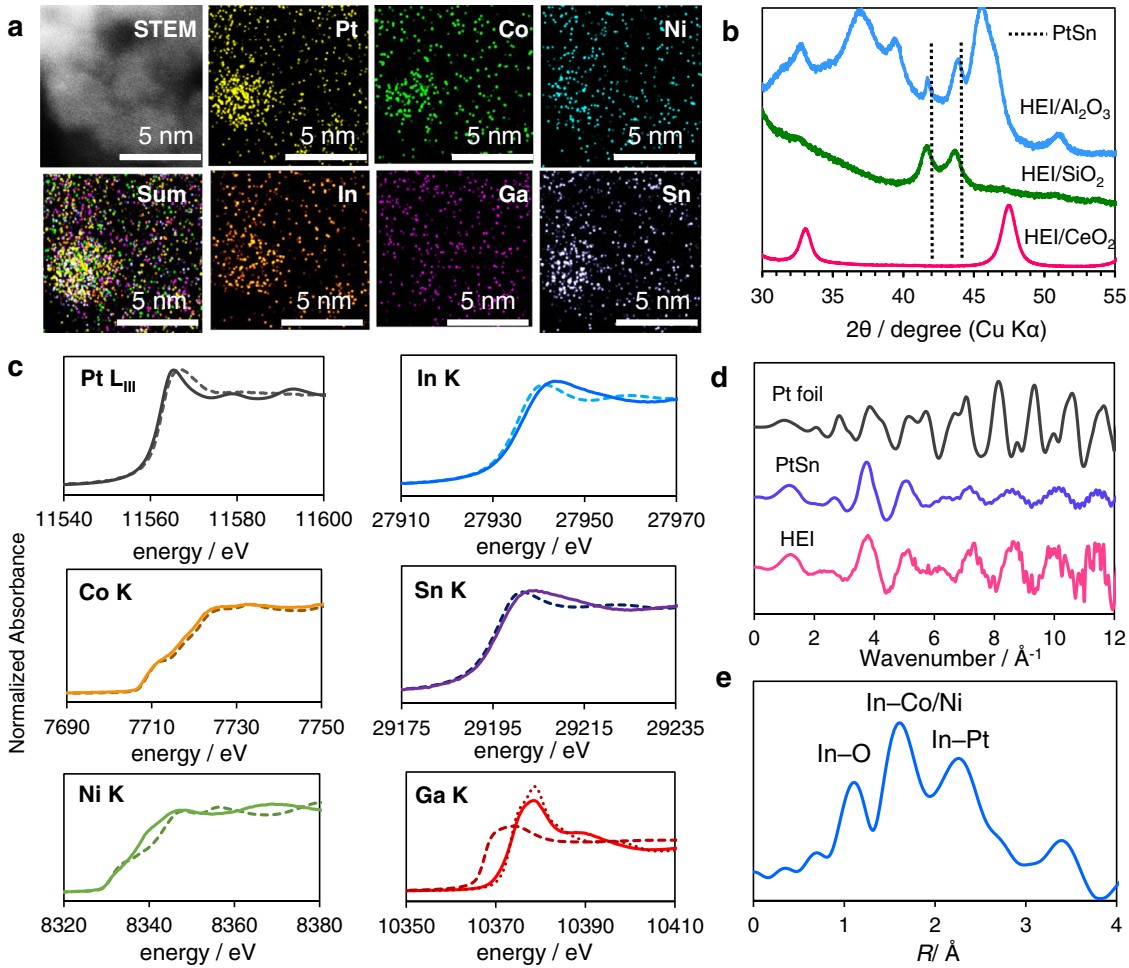

**Fig. 2 | Characterization of the HEI catalyst. a** HAADF-STEM image and the corresponding elemental maps of HEI/CeO$_2$. **b** XRD patterns of HEI supported on Al$_2$O$_3$, SiO$_2$ and CeO$_2$. **c** Pt L$_{III}$, Co K, Ni K, In K, Sn K, and Ga K-edge XANES spectra of the in-situ reduced catalysts (solid lines) and reference foils (dashed lines) or oxides (dotted lined). **d** Pt L$_{III}$ -edge EXAFS oscillations of Pt foil, Pt–Sn and HEI/CeO$_2$. **e** Fourier transform of the In K-edge EXAFS of HEI/CeO$_2$.

conducted extensive curve-fitting analysis on other adsorption edges, assigning the corresponding transition metal–typical metal scatterings (Pt–Sn/In, Co–Sn/In, Ni–Sn/In, Sn–Pt, and Sn–Co/Ni). Although metallic bonds with Ga were suggested, their coordination numbers were small, whereas the coordination number of Ga–O was confirmed, which is consistent with the XANES result. Based on these findings, PtSn-based HEI nanoparticles, i.e., (PtCoNi)(SnInGa), were most likely formed on CeO$_2$, with the Pt and Sn sites being partially replaced primarily with Co/Ni and In/Ga, respectively. Notably, the fraction of metallic Ga in the HEI nanoparticles appears to be lower than suggested by XANES and EXAFS because the Ga species are present primarily as oxides (they do not participate in alloy formation).

**Catalytic performance in CO$_2$-ODP**

The catalytic performance of HEI/CeO$_2$ in CO$_2$-ODP was then tested at 600 °C. Additionally, control catalysts such as Pt/CeO$_2$, Pt–Co–In/CeO$_2$, and PtSn/CeO$_2$ were also tested. Figure 3a–c show the time-course of C$_3$H$_8$ conversion, C$_3$H$_6$ selectivity in hydrocarbons (see Supplementary Fig. 8 for the net C$_3$H$_6$ selectivity, which includes CO formed from hydrocarbons via dry reforming), and CO$_2$ conversions, respectively. With the exception of Pt/CeO$_2$ (72% sel.), all catalysts converted C$_3$H$_8$ by approximately 30% with a high C$_3$H$_6$ selectivity (90–94%). The net C$_3$H$_6$ selectivity was comparable to that in hydrocarbons for PtSn and HEI, whereas it was lower for Pt and Pt–Co–In, indicating that dry reforming of propane was inhibited on the PtSn-based structure. Although Pt,

PtSn, and Pt–Co–In showed rapid deactivation within 5–20 h, HEI/CeO$_2$ mostly retained the initial conversion at least for 30 h. Similar trends were observed for C$_3$H$_6$ selectivity and CO$_2$ conversion, with HEI/CeO$_2$ retaining the highest C$_3$H$_6$ selectivity (90%) and the lowest deactivation rate for 50 h, as well as the highest stability for CO$_2$ conversion. The trends in catalyst stability can be explained roughly by the amount of coke on the catalyst as estimated by temperature-programmed oxidation (TPO, Fig. 3d), where the relative coke amount on HEI was much lower than that on Pt, PtSn, and Pt–Co–In even after 50 h of catalytic run. The coke selectivity, which was calculated by dividing the mole of accumulated coke by the total mole of the converted C$_3$H$_8$ and the carbon number of C$_3$H$_8$, was only 0.001% (Supplementary Table 3), highlighting its outstandingly high coke resistance (material balance was also close to unity, Supplementary Fig. 9). The loss of C$_3$H$_6$ selectivity at the later stage of the reaction for PtSn and Pt–Co–In might be due to the segregation of Pt from the alloy phase because the final selectivity was close to that for Pt. Furthermore, we used CO chemisorption on spent catalysts to evaluate whether the metal dispersion was retained following the reaction. The spent catalyst was regenerated through oxidation–reduction treatment before the CO chemisorption. As shown in Fig. 3e (for details see Supplementary Table 4), the metal dispersion of PtSn and Pt–Co–In decreased significantly following the reaction, implying that nanoparticles sinter irreversibly. On the other hand, HEI retained the original metal dispersion, which demonstrated that

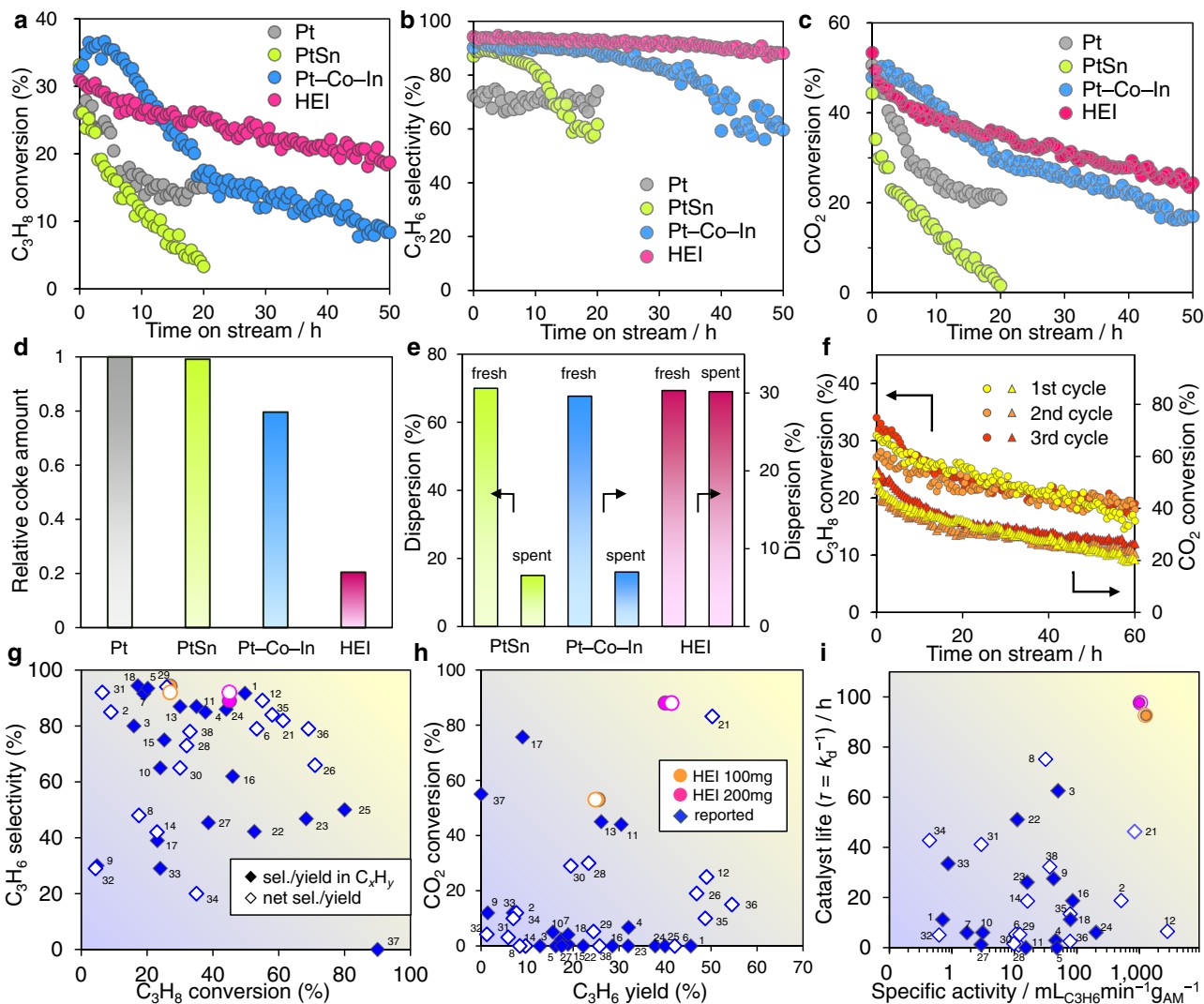

**Fig. 3 | Catalytic performance of HEI/CeO₂ in the CO₂-ODP.** Reaction conditions: catalyst amount, 100 mg; gas feed, $C_3H_8:CO_2:He = 5:5:10$ mL·min⁻¹; temperature, 600 °C. **a–c** Time course of (**a**) $C_3H_8$ conversion, (**b**) $C_3H_6$ selectivity in hydrocarbons, and (**c**) $CO_2$ conversion. **d** Relative coke amount accumulated by $O_2$-TPO of the spent catalysts after the time stream of **a** in CO₂-ODP at 600 °C. **e** Metal dispersions comparison of fresh and spent catalysts. **f** Reusability of HEI/CeO₂ in repeated long catalytic runs with regeneration procedure ($CO_2:He = 5:10$ mLmin⁻¹

for 2 h, then $H_2:He = 5:10$ mLmin⁻¹ for 0.5 h at 600 °C between each cycle). **g–i** Comparison of the catalytic performances with those of reported systems (values indicate references in Supplementary Tables 5, 6): **g** $C_3H_8$ conversion vs. $C_3H_6$ selectivity, (**h**) $C_3H_6$ yield vs $CO_2$ conversion, (**i**) specific activity of $C_3H_6$ production ($mL_{C3H6}$ min⁻¹ $g_{AM(active\ metal)}$⁻¹) vs. mean catalyst life ($\tau = k_d$⁻¹: reciprocal deactivation constant).

no aggregation occurred even under the harsh conditions. Thus, the HEI/CeO₂ catalyst demonstrated remarkably high thermal stability and coke resistance, enabling long-term stability in CO₂-ODP.

We also tested the catalytic performance of other Pt-based binary alloy catalysts (Pt–M/CeO₂: M = Co, Ni, In, and Ga, Supplementary Fig. 10) to confirm the role of constituent metals. Although Pt–Co and Pt–Ni exhibited higher initial $C_3H_8$ and $CO_2$ conversions than Pt, $C_3H_6$ selectivity was quite low (20%–40%) because of a large contribution of dry reforming of propane (DRP) to CO. This suggests that typical metals are necessary to inhibit DRP. However, Pt–Ga/CeO₂ and Pt–In/CeO₂ demonstrated very low $CO_2$ conversion and higher $C_3H_6$ selectivity; thus, Ni and Co are required for $CO_2$ activation as expected. We also investigated the effect of multi-metallization at the Sn site with PtCoNiSn/CeO₂ (Pt: Ni: Co: Sn = 1:1:1:3) and PtCoNiSnIn/CeO₂ (Pt: Ni: Co: Sn = 1:1:1:1.5:1.5), which resulted in lower catalyst stability than HEI/CeO₂, respectively (Supplementary Figs. 11, 12). As a result, the formation of the HEI structure is a key factor in achieving high stability.

Using HEI/Al₂O₃ and HEI/SiO₂ catalysts, the effect of CeO₂ support was also investigated (Supplementary Fig. 13). Although high $C_3H_8$ conversion (50%-60%) and $C_3H_6$ selectivity (> 95%) were obtained at the beginning of the reaction, deactivation occurred rapidly due to coke accumulation within a few hours. This demonstrates that while the PtSn-based HEI has an intrinsically high performance for CO₂-ODP, CeO₂ is also required to maintain long-term stability.

The reusability of HEI/CeO₂ was then tested. The spent catalyst was regenerated by flowing $CO_2$ at 600 °C, followed by $H_2$ reduction. Both $C_3H_8$ and $CO_2$ conversions were fully recovered after the repeated regeneration processes (Fig. 3f). Thus, the HEI/CeO₂ catalyst demonstrated exceptional stability, regenerability, and coke resistance in CO₂-ODP. When the spent PtSn/CeO₂ catalyst underwent the regeneration process including $H_2$ reduction, $C_3H_6$ selectivity was recovered to the original level (Supplementary Fig. 14), indicating that the segregated Pt–$SnO_x$ composite was alloyed again. However, the conversion of $C_3H_8$ and $CO_2$ was not recovered, which is consistent with the increase in the size of nanoparticles by sintering. In terms of

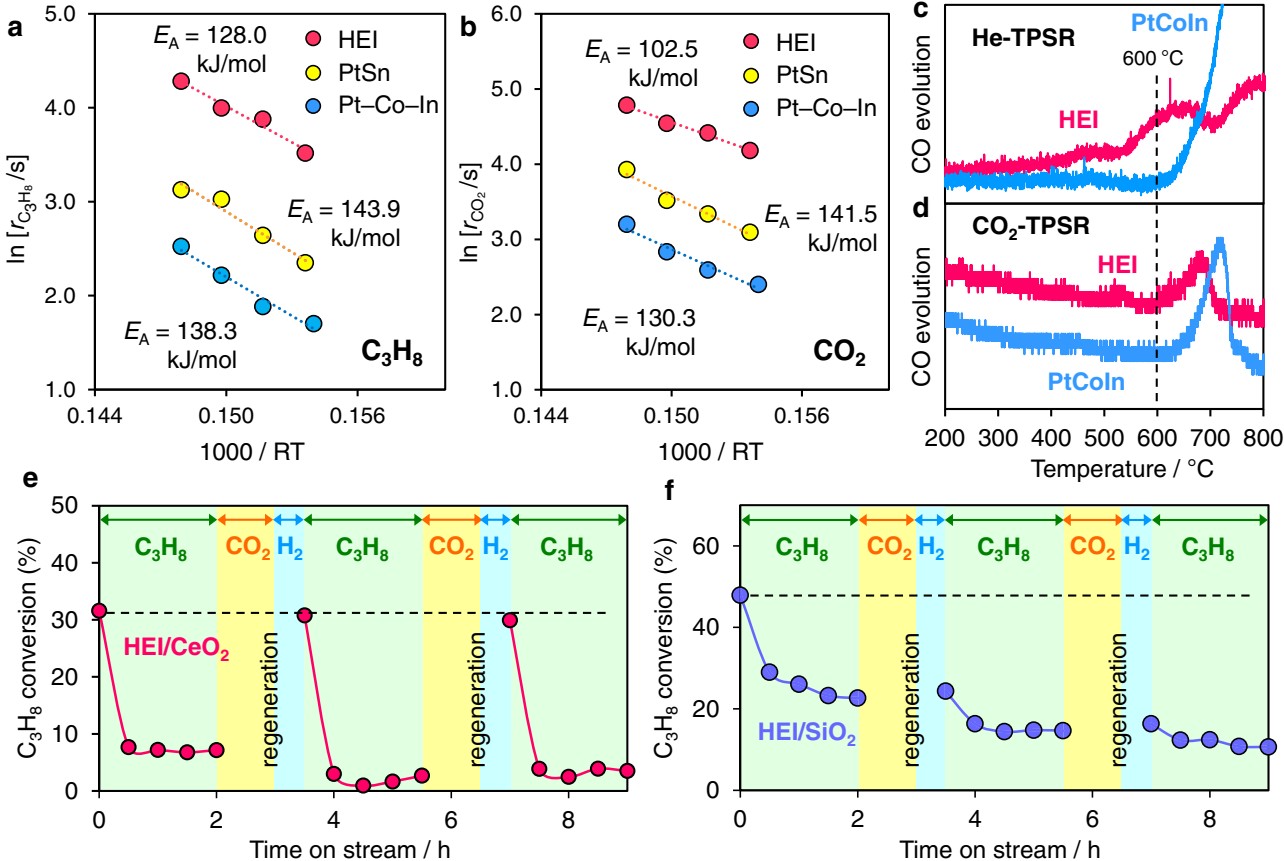

**Fig. 4 | Mechanistic study.** Arrhenius-type plots for **a** $C_3H_8$ and **b** $CO_2$ conversion rates obtained in $CO_2$–ODP on HEI/CeO$_2$. **c** He- and **d** $CO_2$-TPSR on the coked Pt−Co−In/CeO$_2$ and HEI/CeO$_2$ catalysts (used in $CO_2$-ODP for 50 h). CO evolution was quantified by the mass intensity of m/z = 28, respectively. Changes in $C_3H_8$ conversion in the DDP–regeneration cycles over **e** HEI/CeO$_2$ and **f** HEI/SiO$_2$. Reaction conditions: DDP; $C_3H_8$:He = 5:10 mLmin$^{-1}$ (2 h), regeneration; $CO_2$:He = 10:10 mLmin$^{-1}$ (1 h), then $H_2$:He = 5:10 mLmin$^{-1}$ (0.5 h) at 600 °C.

activity, selectivity, stability, reusability, and $CO_2$ utilization efficiency, the catalytic performance of HEI/CeO$_2$ was compared to that of previously reported systems for the $CO_2$-ODP (Fig. 3g–i; see Supplementary Figs. 15, 16 and Supplementary Tables 5, 6 for details with references). Here, two selectivity/yield descriptions (filled: in hydrocarbons, open: including CO via dry reforming) are shown for better comparison with those in literature because the description differs depending on literature. The HEI/CeO$_2$ catalyst demonstrated exceptional catalytic activity, excellent $C_3H_6$ selectivity, high $CO_2$ utilization efficiency, and long-term stability. Notably, the mean catalyst life ($\tau = k_d^{-1}$) was 13 times that of the PtSn/CeO$_2$ catalyst and twice that of the Pt−Co−In/CeO$_2$ catalyst.

**Mechanistic study**

A mechanistic study was performed to better understand the roles of the HEI structure in enhanced catalysis. Arrhenius-type plots were used to compare the apparent activation energy ($E_A^*$) for the PtSn/CeO$_2$, Pt−Co−In/CeO$_2$, and HEI/CeO$_2$. The $E_A$ of $C_3H_8$ dehydrogenation for PtSn/CeO$_2$ was 143.9 kJ·mol$^{-1}$ and Pt−Co−In/CeO$_2$ was 138.3 kJ·mol$^{-1}$, whereas HEI/CeO$_2$ was 128.0 kJ·mol$^{-1}$ (Fig. 4a). A similar trend was also observed for the $E_A$ of $CO_2$ reduction (Fig. 4b, PtSn: 141.5 kJ·mol$^{-1}$, Pt−Co−In: 130.3 kJ·mol$^{-1}$, and HEI: 102.5 kJ·mol$^{-1}$). Thus, the activation of $C_3H_8$ and $CO_2$ are both kinetically promoted in the HEI structure. To investigate the reactivity of coke accumulated on the catalyst surface, we also conducted temperature-programmed surface reactions (TPSRs) on spent Pt−Co−In/CeO$_2$ and HEI/CeO$_2$ catalysts. When the temperature was elevated under flowing He, CO was produced, indicating that coke was combusted by CeO$_2$ lattice oxygen. (He-TPSR,

Fig. 4c). Notably, the combustion temperature for HEI was lower than that of Pt−Co−In, demonstrating that the coke combustion ability of HEI/CeO$_2$ is superior to that of Pt−Co−In/CeO$_2$. This may be explained by the enhanced redox property of CeO$_2$ by multi-metallization (see Supplementary Fig. 17 for details). When TPSR was performed on coked HEI/CeO$_2$ in the presence of $CO_2$ ($CO_2$-TPSR, Fig. 4d), a large amount of CO evolved from 600 °C and was completely combusted at 700 °C. On the other hand, when coked Pt−Co−In/CeO$_2$ was used, CO evolution occurred at higher temperatures (from 650 °C), which is consistent with the trend in $E_A$ of $CO_2$ activation. Based on these results, we concluded that the high coke resistance of HEI/CeO$_2$ originated from the enhancement in the redox property of CeO$_2$ and the $CO_2$ activation ability of the alloy phase by multi-metallization. We further tested the catalytic performance of HEI in DDP as a control experiment without $CO_2$. As shown in Fig. 4e, rapid deactivation occurred within 0.5 h even for HEI/CeO$_2$, indicating that simultaneous $CO_2$ supply is necessary for the continuous coke combustion. This is because the oxygen vacancy of CeO$_2$ must be refilled for the continuous combustion. In this regard, we previously confirmed by $H_2$-TPR and $CO_2$-TPO that the oxygen atoms derived from $CO_2$ refilled the oxygen vacancy of CeO$_2$[14]. Then, $C_3H_8$ conversion was fully recovered when HIE/CeO$_2$ was regenerated under flowing $CO_2$ and the subsequent $H_2$ reduction, demonstrating that coke accumulated on the catalyst can be completely removed. Conversely, the catalytic activity of HEI/SiO$_2$ (Fig. 4f) and HEI/Al$_2$O$_3$ (Supplementary Fig. 18) was not fully recovered after the regeneration process. Therefore, the oxygen releasing ability of CeO$_2$ is essential for the coke combustion and the high regenerability. HEI/SiO$_2$ showed slower deactivation rate than

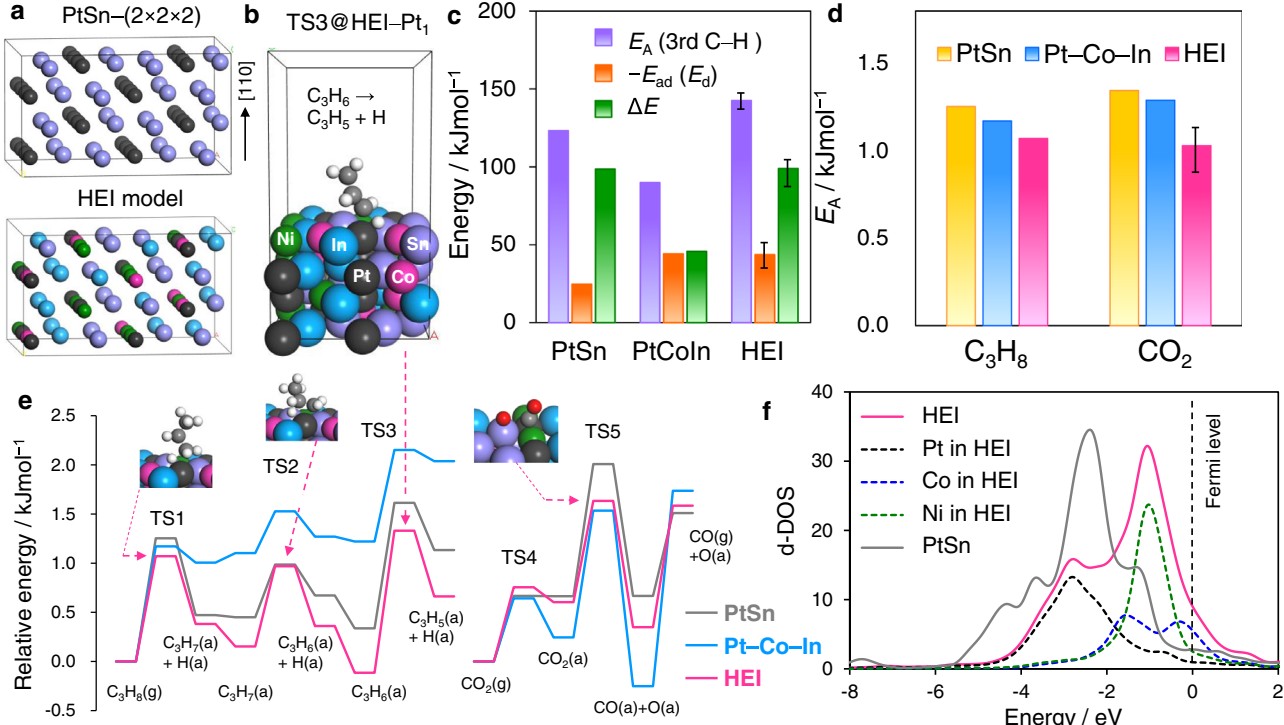

**Fig. 5 | DFT calculations. a** Model structures of PtSn and the PtSn-based HEI for DFT calculations. **b** An example of the HEI slab model for $CO_2$-ODP for the over dehydrogenation: $C_3H_6 \rightarrow C_3H_5$+H at a $Pt_1$-HEI site on (110) surface (HEI(001):B3). **c** Comparison of $E_A$ of the third C–H activation (overdehydrogenation of $C_3H_6$), $-E_{ad}$ ($E_d$) of $C_3H_6$, and $\Delta E$ (= $E_A + E_{ad}$) on the surface of PtSn, Pt–Co–In, and HEI (average). **d** $E_A$ of propane dehydrogenation and $CO_2$ activation. For propane dehydrogenation, $E_A$ of the first C–H scission was shown. **e** Energy diagrams of propane dehydrogenation and $CO_2$ reduction on the surface of PtSn, Pt–Co–In and HEI. **f** d-DOS of the surface transition metals on HEI(004):B and PtSn(110).

HEI/$CeO_2$, which may be due to much higher specific surface area of $SiO_2$ than $CeO_2$.

## DFT calculation

Finally, DFT calculations were conducted to better understand the role of the HEI phase in the $CO_2$-ODP's high catalytic performance. As the parent structure of HEI, a PtSn–(2 × 2 × 2) supercell was constructed, and then the Pt and Sn sites were partially and randomly replaced with Ni/Co and In, respectively (Fig. 5a, see method paragraph and Supplementary Table 7 for further details in modeling). Because of the low concentration mentioned above, Ga was not included in this model for simplicity. Two (110) surfaces (corresponding to the supercell's (004) layer) were cleaved for surface slab models (Supplementary Fig. 19). Adsorption ($E_{ad}$) and C–H activation energies ($E_A$) of propylene were calculated using eight different adsorption sites and conformations (Fig. 5b, see Supplementary Figs. 20–24 for details). $E_{ad}$ and $E_A$ have narrow ranges ($E_{ad}$: −51 ~ −35 kJmol⁻¹, $E_A$: 137 ~ 147 kJmol⁻¹), suggesting that reactivity is not strongly dependent on local structure and elemental distribution. Figure 5c compares the $E_A$ and $E_{ad}$ for PtSn, Pt–Co–In, and HEI. We also demonstrated $\Delta E$ ($\Delta E = E_A + E_{ad} = E_A − E_d$ (propylene desorption energy)), which is widely used as a scale that reflects propylene selectivity. The $\Delta E$ of HEI (98.9 kJmol⁻¹) was much higher than that of Pt–Co–In (45.7 kJmol⁻¹) and indicates that propylene desorption is significantly enhanced, which is consistent with the experimental trend in $C_3H_6$ selectivity (90% and 60% at 40 hours, respectively). When compared to Pt–Co–In and PtSn, HEI demonstrated much higher $E_A$, indicating that isolating Pt effectively inhibits the third C–H activation, which causes side effects. This property significantly reduces the formation of coke as a result of side reactions. The $E_A$ of propane to propylene and $CO_2$ reduction was also calculated on each of the three surfaces (see Supplementary Figs. 20–29 for details). The summary of $E_A$ and the corresponding energy diagrams

are shown in Fig. 4d, e, respectively. HEI showed lower $E_A$ of propane dehydrogenation than PtSn and Pt–Co–In, indicating that the desired C–H activation was enhanced by multi-metallization while the undesired was inhibited. The order of $E_A$ (HEI < Pt–Co–In < PtSn) was consistent with that of the experimental apparent $E_A$ shown in Fig. 4a. For $CO_2$ reduction, $CO_2$ is first chemisorbed on the metallic surface to form a bidentate $sp^2$-like conformation, then C–O cleavage occurs to generate CO and O (Fig. 5e and Supplementary Figs. 25–28). Although the chemisorption step showed moderate energy barriers (~ 0.7 eV), they were much lower than those of the $CO_2$ activation step (1.0-1.3 eV), indicating that $CO_2$ activation is the rate-determining step (RDS) in $CO_2$ reduction. In this regard, we recently revealed by microkinetic modeling that $CO_2$ activation was the RDS in $CO_2$-ODP over Pt–Co–In. Among the three surfaces, PtSn had the highest $E_A$ of $CO_2$ activation (Fig. 5d). This is consistent with the low $CO_2$ conversion rate and is most likely due to a lack of 3d transition metals. The $E_A$ of HEI was exhibited to be lower than that of Pt–Co–In, which corresponded to the experimental order of apparent activation energies shown in Fig. 4a. We also considered the effect the electronic structure changes upon multi-metallization on the catalytic activity. Figure 5f shows the density of states projected on d orbitals (d-DOS) of the surface transition metals on HEI(004):B and PtSn(110). PtSn had low d-DOS near the Fermi level due to alloying with Sn, which may have resulted in the high $E_A$. Conversely, HEI showed an intense peak near the Fermi level owing to the Ni and Co 3d states. Therefore, the d band was significantly upshifted near the Fermi level by doping Ni and Co. The enhanced activity in propane dehydrogenation and $CO_2$ reduction can be explained by the modification of the d band by the introduction of Ni and Co. This demonstrates our catalyst design concept mentioned in the introduction paragraph (Fig. 1).

As a result, the experimental trends in $C_3H_8$ activation, $C_3H_6$ selectivity, $CO_2$ conversion, and stability were reproduced by our

DFT calculation. The enhanced $CO_2$ activation capability improves oxygen supply for coke combustion. Thus, the HEI phase is capable of inhibiting coke formation while also promoting coke combustion, which enhances coke resistance and the long-term stability of the catalyst.

## Discussion

In summary, we designed and prepared a PtSn-based HEI catalyst in the $CO_2$-ODP using $CeO_2$ as a highly coke resistant and thermally stable catalyst. The Pt and Sn sites of intermetallic PtSn were partially substituted with Ni/Co and In/Ga, resulting in nanoparticles with a HEI structure of (PtCoNi)(SnInGa). The multi-metallization sufficiently isolates Pt atoms, preventing propylene decomposition and coke formation. The incorporation of Ni and Co significantly enhanced the ability to activate $CO_2$. Doping In(Ga) into the Sn site improved catalyst stability, probably due to the enhanced entropy effect. The combination of these abilities significantly enhances coke resistance and catalyst lifetime. Furthermore, because of the increased thermal stability, multi-metallization also prevents nanoparticle sintering. The HEI/$CeO_2$ catalyst shows 2- and 1.5-times higher catalyst life and specific activity than those of the best catalyst ever reported. Moreover, the catalyst can be reused by performing a simple regeneration procedure with $CO_2$ alone without any loss of original performance. This work not only demonstrated outstanding catalytic performance, but it also opened up a new window of catalyst design concepts based on HEI. The obtained insights and technology will contribute to carbon-neutralization of industrial processes for light alkane conversion.

## Methods

### Catalyst preparation

Pt/$CeO_2$, Pt–Co–In/$CeO_2$, HEI/$CeO_2$, Pt–Sn/$CeO_2$, Pt–Co–In/$CeO_2$, quinary, HEA and HEI/$CeO_2$ (Pt: 1 wt%) were prepared by a conventional impregnation method using $H_2PtCl_6$ (aqueous solution, Kojima Chemicals, Pt 3.71 wt%), $In(NO_3)_3\cdot3H_2O$, $Co(NO_3)_2\cdot6H_2O$, $Ni(NO_3)_2\cdot6H_2O$, $Ga(NO_3)_3\cdot6H_2O$ and $SnCl_2$ as metal precursors. The $CeO_2$ support (JRC-CEO-2, $S_{BET} = 123.1\ m^2g^{-1}$) was added to a vigorously stirred aqueous solution (50 ml $H_2O$ per gram of $CeO_2$) containing Pt and the corresponding second and/or third metal precursor(s) (Pt:Co:Ni:Sn:In:Ga=1:1:1:1:1:1, Pt:Ni=1:1, Pt:Co =1:1, Pt:In =1:1, Pt:Ga =1:1, Pt:Sn = 1:1, and Pt:Co:In =1:1:2), followed by stirring for 3 h at room temperature. The mixture was dried under a reduced pressure at 50 °C using a rotary evaporator, followed by calcination under flowing air at 500 °C for 1 h and reduction under flowing $H_2$ (50 ml/min) at 600 °C for 1 h. HEI/ $Al_2O_3$ catalysts ($Al_2O_3$ prepared by calcination of boehmite [γ-AlOOH, supplied by SASOL chemicals] at 900 °C for 3 h, γ-phase), Pt:Co:Ni:Sn:In:Ga=1:1:1:1:1:1, Pt: 1 wt%) was also prepared by the same method mentioned above. HEI/$SiO_2$ was synthesized by the pore-filling co-impregnation method, which can deposit all the metal components on the $SiO_2$ support without loss[17].

### Catalytic test

Under atmospheric pressure, $CO_2$-ODP was performed in a quartz fixed-bed reactor with a 6 mm internal diameter. Prior to the catalytic reactions, the catalyst (0.10 g) diluted with sea sand (0.90 g, Miyazaki Chemical, 99.9%) was treated with flowing hydrogen (10 mL/min) at 600 °C for 0.5 h. The catalysts were then evaluated by feeding a reactant gas mixture ($C_3H_8$: $CO_2$: He = 1:1:2, a total flow rate of 20 mL/min) through them. The gas phase was analyzed and quantified using a downstream-equipped online thermal conductivity detection gas chromatograph (Shimadzu GC-8A, column: Gaskuropack 54, 80/100 SUS 2 m × 3 mm I.D.). $C_3H_8$, $C_3H_6$, $C_2H_4$, $C_2H_6$, $CH_4$, CO, and $CO_2$ could be separated under the following condition (see Supplementary Fig. 30 for a GC chart): carrier gas; He 30 mL min$^{-1}$, column temperature; 50 °C (constant). The conversions of $C_3H_8$ and $CO_2$ were defined as follows:

$$C_3H_8\ \text{conversion}: X_{C_3H_8}(\%) = \frac{F_{C_3H_8}^{in} - F_{C_3H_8}^{out}}{F_{C_3H_8}^{in}} \times 100 \quad (1)$$

$$CO_2\ \text{conversion}: X_{CO_2}(\%) = \frac{F_{CO_2}^{in} - F_{CO_2}^{out}}{F_{CO_2}^{in}} \times 100 \quad (2)$$

where, $F_x^{in}$ and $F_x^{out}$ indicate the $x$ (mL min$^{-1}$), inlet and outlet flow rates of respectively.

In this reaction, CO can be formed from $C_xH_y$ by dry reforming as well as from $CO_2$ via the reverse water gas shift reaction, which was distinguished as follows:

$$CO\ \text{formed from}\ CO_2: F_{CO}^{CO_2} = F_{CO_2}^{in} - F_{CO_2}^{out}(\text{mL} \cdot \text{min}^{-1}) \quad (3)$$

$$CO\ \text{formed from}\ C_xH_y: F_{CO}^{C_xH_y} = F_{CO}^{out} - F_{CO}^{CO_2}(\text{mL} \cdot \text{min}^{-1}) \quad (4)$$

Then, two different expressions of $C_3H_6$ selectivity and yield were defined in order to compare them to those reported in the literature.

$$C_3H_6\ \text{sel.in}\ C_xH_y: S_{C_3H_6}^{C_xH_y}(\%) = \frac{F_{C_3H_6}^{out}}{F_{C_3H_6}^{out} + \frac{2}{3}F_{C_2H_6}^{out} + \frac{2}{3}F_{C_2H_4}^{out} + \frac{1}{3}F_{CH_4}^{out}} \times 100 \quad (5)$$

$$\text{net}\ C_3H_6\ \text{sel.}: S_{C_3H_6}(\%) = \frac{F_{C_3H_6}^{out}}{F_{C_3H_6}^{out} + \frac{2}{3}F_{C_2H_6}^{out} + \frac{2}{3}F_{C_2H_4}^{out} + \frac{1}{3}F_{CH_4}^{out} + \frac{1}{3}F_{CO}^{C_xH_y}} \times 100 \quad (6)$$

$$C_3H_6\ \text{yield in}\ C_xH_y: Y_{C_3H_6}^{C_xH_y}(\%) = \frac{X_{C_3H_8} \cdot S_{C_3H_6}^{C_xH_y}}{100} \quad (7)$$

$$\text{net}\ C_3H_6\ \text{yield}: Y_{C_3H_6}(\%) = \frac{X_{C_3H_8} \cdot S_{C_3H_6}}{100} \quad (8)$$

Material balance was considered using the following scales:

$$\text{material blance in}\ C_xH_y = \frac{F_{C_3H_8}^{out} + F_{C_3H_6}^{out} + \frac{2}{3}F_{C_2H_6}^{out} + \frac{2}{3}F_{C_2H_4}^{out} + \frac{1}{3}F_{CH_4}^{out}}{F_{C_3H_8}^{in}} \times 100 \quad (9)$$

$$\text{material blance in}\ CO_x = \frac{F_{CO_2}^{out} + F_{CO}^{out}}{F_{CO_2}^{in}} \times 100 \quad (10)$$

total material balance

$$= \frac{F_{C_3H_8}^{out} + F_{C_3H_6}^{out} + \frac{2}{3}F_{C_2H_6}^{out} + \frac{2}{3}F_{C_2H_4}^{out} + \frac{1}{3}F_{CH_4}^{out} + F_{CO_2}^{out} + \frac{1}{3}F_{CO}^{C_xH_y} + F_{CO}^{CO_2}}{F_{C_3H_8}^{in} + F_{CO_2}^{in}} \times 100 \quad (11)$$

The deactivation constant, mean catalyst life, $CO_2$ utilization efficiency, turnover frequency (TOF), average coke selectivity, and turnover number of coke were defined as follows.

$$\text{Deactivation constant}: k_d = \left\{ \ln\left(\frac{1 - X_{C_3H_8}^f}{X_{C_3H_8}^f}\right) - \ln\left(\frac{1 - X_{C_3H_8}^i}{X_{C_3H_8}^i}\right) \right\} \left(t^f - t^i\right)^{-1} \quad (12)$$

$$\text{Mean catalyst life}: \tau = \frac{1}{k_d} \quad (13)$$

$$CO_2 \text{ utilization efficiency } (\%) = X_{CO_2}\left(1 - \frac{|X_{CO_2} - Y_{C_3H_6}|}{X_{CO_2} + Y_{C_3H_6}}\right) \quad (14)$$

$$TOF_{C_3H_8/CO_2}(\text{site}^{-1}\text{min}^{-1}) = \frac{M_{C_3H_8/CO_2}^{in} \times X_{C_3H_8/CO_2}^{i}}{n_{Pt+Co+Ni} \times D} \quad (15)$$

$$\text{average coke selectivity}: S_C \,(\%) = \frac{\frac{1}{3}n_C^t}{\int_0^t \left(F_{C_3H_8}^{in} - F_{C_3H_8}^{out}\right)dt} \times 100 \quad (16)$$

$$TON_{coke}\left(\text{site}^{-1}\right) = \frac{n_C^t}{n_{Pt+Co+Ni} \times D} \quad (17)$$

$$\text{specific activity}(\text{mL}_{C_3H_6}\,\text{min}^{-1}\text{g}_{AM}^{-1}) = \frac{F_{C_3H_8}^{in} \times Y_{C_3H_6}}{W_{AM}} \quad (18)$$

where, $X_{C_3H_8}^{i}$ and $X_{C_3H_8}^{f}$ indicate the initial ($t^i$: 0.5 h) and final ($t^f$: 50 h) $X_{C_3H_8}$, respectively. The $CO_2$ utilization efficiency was used as the scale of how much $CO_2$ is converted ($X_{CO_2}$) and how close to the 1:1 stoichiometry the consumption of $CO_2$ and the formation of $C_3H_6$ (the latter term: maximum (unity) at the 1:1 stoichiometry) are. The $CO_2$ utilization efficiency becomes high when $CO_2$ conversion is high and when the stoichiometry of $CO_2$ conversion and $C_3H_6$ formation is close to unity, i.e., $CO_2$ is solely used for $CO_2$-ODP. Conversely, it becomes low when $CO_2$ conversion is much higher than $C_3H_6$ yield or when excess $CO_2$ is used. $M_{C_3H_8/CO_2}^{in}$ and $M_{C_3H_8/CO_2}^{out}$ are the inlet and outlet molar flow rates (mol/min) of $C_3H_8$ or $CO_2$, respectively. $n_{Pt+Co+Ni}$ and $D$ corresponds to the mole of active metal in the catalyst and its dispersion estimated by CO chemisorption, respectively. We confirmed by DFT calculation that CO can strongly adsorb on Pt, Co, and Ni sites (Supplementary Fig. 31). $n_C^t$ indicates the mole of coke accumulated at a time on a stream of $t$ ($x = C$). $W_{AM}$ is the weight (g) of the active metal component included in the catalyst. The second term in the parenthesis of Eq. (14) represents the degree of deviation between $X_{CO_2}$ and $Y_{C_3H_6}$. All catalytic tests for kinetic analysis were performed under different conditions by adjusting the amount of catalyst used, where the reactant conversion was lower than 15% (typically 5%~10%).

**Characterization**

The crystal structure of the prepared catalyst was investigated using powder XRD on a Rigaku MiniFlex II/AP diffractometer with Cu Kα radiation. A JEOL JEM-ARM200 M microscope equipped with an EDX analyzer was used for high-angle annular dark field scanning transmission electron microscopy (HAADF-STEM) (EX24221M1G5T). The STEM analysis was performed at a 200 kV accelerating voltage. To prepare the TEM specimen, all samples were sonicated in ethanol and then dispersed on a Mo grid supported by an ultrathin carbon film.

A TPO experiment was performed using BELCAT II (MicrotracBEL) to determine the amount of coke deposited on spent catalysts after 20 and 50 hours of $CO_2$-ODP at 600 °C (0.1 g of the catalyst with 0.9 g of quartz sand). The spent catalyst was placed in a quartz tube reactor and treated with flowing He (30 mL/min) at 150 °C for 30 minutes before cooling to room temperature. While $O_2$/He (50%, 40 mL/min) was flowing, the catalyst bed temperature was increased (40 °C–900 °C, ramping rate: 5 °C min$^{-1}$). An online mass spectrometer was used to determine the amount of $CO_2$ in the outlet gas. $CO_2$- and He-TPSR experiments were performed for coked catalysts in a similar fashion using the flow of $CO_2$/He (10%, 30 mL/min) and pure He (30 mL/min). The dispersion of Pt and Co in the catalysts (the percentage of exposed Pt + Co to the total amount of Pt + Co) was measured by the chemisorption of CO at room temperature. Prior to chemisorption, the catalyst (40 mg) was treated with 5% $H_2$/Ar

(30 mL/min) at 600 °C for 0.5 hours, followed by cooling to ca –110 °C by liquid nitrogen with an He purge (30 mL/min). Then, introduced a pulse of 10% CO/He into the reactor and quantified the CO that passed through the catalyst bed using a TCD detector. This pulse measurement was repeated until no more CO was adsorbed. The dispersion was estimated assuming a 1:1 stoichiometry of CO adsorption on Pt and/or Co. In our ionization condition, the signal intensity of m/z = 28 is only about 6% of that of m/z = 44 when $CO_2$ alone is flowed without catalyst as shown below. However, for He-TPSR, the signal intensity of m/z = 28 is comparable to or higher than that of m/z = 44, indicating that the contribution of $CO_2$ to the signal of m/z = 28 is negligible. For $CO_2$-TPSR, the peak feature in the signal of m/z = 28 was not observed in that of m/z = 44, which purely indicates the evolution of CO.

XAFS spectra of the prepared catalysts were collected at the BL01B14 beamline of SPring-8, Japan Synchrotron Radiation Research Institute (JASRI) using Si(111) (for Co K-, Ni K-, Ga K-, and Pt L$_{III}$-edges) and Si(311) (for Sn K- and In K-edge) double-crystals as monochromators. Prior to the measurement, the catalyst was pelletized (ca. 150 mg with a diameter of 10 mm) and pretreated by $H_2$/$N_2$ (20%, 40 mL/min) at 600 °C for 0.5 hours in an in-situ quartz cell, followed by cooling to room temperature with an $N_2$ purge (32 mL/min). At room temperature, the XAFS spectra were recorded in transmission (Sn K- and In K-edge) and fluorescence (Pt L$_{III}$-, Ga K-, Ni K-, and Co K-edge: using a 19-element Ge solid-state detector) modes. The XAFS spectra were analyzed using the Athena and Artemis software versions 0.9.25, which are included in the Demeter package. The back-scattering amplitude and phase shift functions were calculated using FEFF8[32]. The R-factor ($R^2$) for curve-fitting is defined as follows:

$$R^2 = \sum_i \left\{k^3\chi_i^{exp}(k) - k^3\chi_i^{fit}(k)\right\}^2 \Big/ \sum_i \left\{k^3\chi_i^{exp}(k)\right\} \quad (19)$$

**Computational details**

Periodic DFT calculations were performed using the CASTEP code[33] with Vanderbilt-type ultrasoft pseudopotentials as well as the revised version of Perdew–Burke–Ernzerhof exchange–correlation functional based on the generalized gradient approximation[34]. At a kinetic energy of 360 eV, the plane-wave basis set was truncated. A 0.1 eV Fermi smearing was utilized. The Tkatchenko–Scheffler method was used to analyze dispersion correlations with a scaling coefficient of $s_R = 0.94$ and a damping parameter of $d = 20$[35]. The reciprocal space was sampled using a $k$-point mesh with a spacing of typically 0.04 Å$^{-1}$, as generated by the Monkhorst–Pack scheme[36]. Geometry optimizations and transition state (TS) searches were performed on supercell structures using periodic boundary conditions. The surfaces were modeled using metallic slabs with a thickness of four atomic layers with 13 Å of vacuum spacing. We chose Pt–Sn(110) as the most stable surface, which has the highest surface atom density and diffraction intensity. Geometry optimizations were performed using the Broyden–Fletcher–Goldfarb–Shanno (BFGS) algorithm[37]. The unit cell size of the bulk material (Pt–Sn, Pt–Co–In and HEI) was firstly optimized, followed by modeling the slab structure and surface relaxation with the size of the supercell fixed. The convergence criteria for structure optimization and energy calculation were set to (a) an SCF tolerance of $1.0 \times 10^{-6}$ eV per atom, (b) an energy tolerance of $1.0 \times 10^{-5}$ eV per atom, (c) a maximum force tolerance of 0.05 eV Å$^{-1}$, and (d) a maximum displacement tolerance of $1.0 \times 10^{-3}$ Å.

As the parent structure of HEI, a PtSn–(2 × 2 × 2) supercell was constructed so that the (110) plane of PtSn corresponded to the (001) plane of the supercell. The distribution of each constituent metal in HEI was determined with the following restrictions: (1) Pt/Co/Ni occupy the Pt sites while Sn/In do the Sn sites (Ga was not included in this model for simplicity because of the low concentration mentioned

above), (2) Pt atoms do not neighbor each other on all the (110) layers, and (3) the number of each metal in each (110) layer is fixed to 2 or 3 (Pt/Ni/Co) and 4 (Sn/In). The distribution of each atom was determined using random numbers. Then, we generated ten different bulk ($2 \times 2 \times 2$) unit cells, of which energy difference was < 0.5 eV, indicating that the distribution does not strongly influence the stability of bulk structure. Among them, we chose the most stable structure as an energetically likely model as shown in Supplementary Table 7 and Supplementary Fig. 19.

The adsorption energy was defined as follows: $E_{ad} = E_{A-S} - (E_S + E_A)$, where $E_{A-S}$ is the energy of the slab together with the adsorbate, $E_A$ is the total energy of the free adsorbate, and $E_S$ is the total energy of the bare slab. The adsorption energy for an oxygen-preadsorbed slab was calculated using $E_{SH}$, which is the total energy of the oxygen-adsorbed slab, instead of using $E_S$. The d band center ($\varepsilon_d$) was defined as the average energy of the occupied d band relative to the Fermi level as follows:

$$\varepsilon_d = \int_{-\infty}^{0} E\rho_d(E)dE \Big/ \int_{-\infty}^{0} \rho_d(E)\,dE \qquad (20)$$

where $\rho_d$ is the density of states projected to d orbitals.

The transition state (TS) search was performed using the complete linear synchronous transit/quadratic synchronous transit (LST/QST) method[38,39]. Linear synchronous transit maximization was performed, followed by energy minimization in the directions conjugating to the reaction pathway. The approximated TS was used to perform QST maximization with conjugate gradient minimization refinements. This cycle was repeated until a stationary point was found. Convergence criterion for the TS calculations were set to root-mean-square forces on an atom tolerance of 0.05 eV Å$^{-1}$.

## Data availability
The data that support the findings of this study are available from the corresponding author upon reasonable request.

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

## Acknowledgements

This work was supported by JSPS KAKENHI (Grant Numbers 17H04965, 20H02517, 22J11748), JST CREST (JPMJCR17J3), JST PRESTO (JPMJPR19T7), and JST SPRING (JPMJSP2119). The XAFS analysis was performed with the approval of JASRI (Nos. 2021A1541, 2021A1571, 2021B1795, and 2021B1962). We appreciate the technical staff of the faculty of engineering at Hokkaido University and the Research Institute for Electronic Science at Hokkaido University for helping with HAADF-STEM observation. Computation time was provided by the super-computer systems in the Institute for Chemical Research at Kyoto University.

## Author contributions

S.F. and F.X. design the research and co-wrote the manuscript in discussion. F.X. performed all the experimental works. J.M. contributed to catalytic performance experiment. S.F. conducted all the computational studies. S.F., F.X., J.M., and K.S. discussed the data and commented on the manuscript.

## Competing interests

The authors declare no competing interests.
