## [Peer Review File · Nature Communications]

Title: High-entropy intermetallics on ceria as efficient catalysts for the oxidative dehydrogenation of propane using CO₂REVIEWER COMMENTS

Reviewer #1 (Remarks to the Author):

This manuscript studies CO₂-ODP catalyzed by PtCoNiInGaSn/CeO₂ with a high-entropy intermetallic structure. Very impressively, highly dispersed PtCoNiInGaSn over CeO₂ is obtained via the simple aqueous impregnation by using chloride or nitrate salts as the metal precursors. Moreover, a high space-time yield of propene around 40 mL C₃H₆·min⁻¹·g_{cat}⁻¹ (Figs. 2i and S12) and a relatively high stability are obtained for CO₂-ODP over the PtCoNiInGaSn/CeO₂ catalyst. These merits make the developed catalyst very promising for CO₂-ODP, which is worthy to be published. However, some key issues must be clarified before any publications as commented as follows.

1. HEI properties.

1.1 Following the authors' definition stated in the manuscript, "HEIs are multi-metallic alloys with five or more elements and specific crystal structures originating from the parent binary intermetallics (Scheme 1)", what is the crystal structure of PtCoNiInGaSn over the studied oxide supports? What is the crystal structure of HEI over Al₂O₃ or SiO₂ with the intense XRD peaks at 41.8 and 44.1°?

1.2 The characterization results confirm that Ga over HEI/CeO₂ is mostly oxidized states rather than the metallic state. In this case, can PtCoNiInGaSn still be HEI? Cannot PtCoNiInSn be formed as HEI? If it can, what will happen for both the crystal structure and the catalytic performance?

1.3 As shown in Fig. 1b, two intense peaks for HEI over Al₂O₃ or SiO₂ are observed at 41.8 and 44.1° while no XRD diffractions are detected for HEI over CeO₂. The authors explain this as that the mean particle size of the metallic species over HEI/CeO₂ is significantly too small to be detectable by XRD. This explanation contradicts the results of the CO chemisorption. According to the dispersion shown in Fig. 2e and Table S4, the particle sizes must be similar for the metals over Pt-Co-In/CeO₂, HEI/CeO₂, and HEI/Al₂O₃ with almost the same metal dispersion of ~30%. Due to the much smaller metal dispersion of 19.7% for HEI/SiO₂, a much larger particle size than that of HEI/CeO₂ or HEI/Al₂O₃ is reasonably expected. The particle size based on CO chemisorption must be calculated and discussed reasonably.

2. Experimental

2.1 Product analysis. "The gas phase was analyzed and quantified using a downstream-equipped online thermal conductivity detection gas chromatograph (Shimadzu GC-8A, 300 column: Gaskuropack 54). C₃H₆, C₂H₄, C₂H₆, CH₄, CO, CO₂, H₂O, and H₂ were detected as reaction products in the outlet gas of all catalysts." Are all of these listed species analyzed by GC with the TCD detector? Can these species be well separated with the GC column? What are the parameters of the GC column? How is the unconverted C₃H₈ analyzed?

2.2. Equations. There are 19 equations for defining conversions, selectivity, TOF, etc. However, many of the results such as the carbon balance, the oxygen balance, and TOF are not provided and discussed. What is the purpose for these definitions? How are the outlet flow rates of different species determined? measured or calculated?

2.3 TPSR. What is m/z used for monitoring CO₂? Is CO simultaneously monitored with the Mass Spectrometer? Is there no CO produced during He-TPSR?

3. Catalytic results.

3.1 The asymmetric conversions of C₃H₈ and CO₂. According to the stoichiometry, the conversions of C₃H₈ and CO₂ for the CO₂-ODP reaction should be equal when the feed molar ratio of C₃H₈ to CO₂ of 1 is applied during the experiments. As shown in Figs. 2a, 2c, 2f, and S8-10, the CO₂ conversion is significantly higher than that of C₃H₈ over the optimal HEI/CeO₂ catalyst, especially in the case of Fig. S10, indicating the severe side reactions. However, the C₃H₆ selectivity of larger than 90% is still obtained (Figs. 2a, 2c, 2f, and S8-10). Why? What are the experimental errors? What is the carbon balance of the experiments? These contradictory results must be carefully checked together with the experimental details missed in the manuscript, which have already commented.

3.2 The caption for b and c of Fig. 3 is interchanged. Moreover, the description of temperatures for CO₂-TPSR over the coked Pt-Co-In/CeO₂ and HEI/CeO₂ catalysts is not matched with Fig. 3c.

3.3 The specific activity in Fig. 2i. In fact, it is the space-time yield of propene. I cannot find how around 40 mL C₃H₆·min⁻¹·gcat⁻¹ (Figs. 2i and S12) are obtained. The results must be carefully checked and confirmed. What is the time on stream for the calculation? Which experimental results are used for the calculation?

3.4 “Thus, the HEI phase is capable of inhibiting coke formation while also promoting coke combustion,-”. Following the conclusion, the promoted coke combustion will decrease the C₃H₆ selectivity. Why is it still high over the HEI/CeO₂ catalyst?

4. Title. What is the meaning of auto-regenerable catalyst? How is it reflected from the results?

5. References. In fact, quantitative works over different catalysts have been reported for CO₂-ODP, especially in recent years. However, it is strange that the authors cite only some purposefully selected works before 2010 both in the references and the comparison in Tables S5 and S6. On the contrary, the authors' many recent works including PDH are cited without any further comparison/discussion in the manuscript. Thus, it is unfair that the important works in recent 5 or 10 years and the very recent review work (ACS Catal., 2021, 11: 2182 – 2234) are not cited, and compared.

Reviewer #2 (Remarks to the Author):

Xing et al. discuss the performance of a multimetallic high entropy intermetallic (HEI) catalyst on ceria support (and variants of this) for CO₂-oxidative dehydrogenation of propane. The catalyst seems active, selective, and stable. The authors compare the proposed HEI catalyst with various alloys and support combinations to show that the reported catalyst is optimal in terms of selectively producing propylene and generating very little coke. While the work clearly shows an interesting class of materials, the experimental and computational analysis of the catalyst in terms of the active sites and reaction mechanism is incomplete or not well discussed. Specific comments are given below

1. How is the turnover frequency calculated? This reviewer guesses it is possibly using the reported CO

chemisorption data. But can chemisorption effectively count sites for the HEI catalyst? It may only bind to certain Pt sites or ensembles (or it may bind to non-Pt ensembles too where the reaction may not take place). A DFT calculation of where CO can bind would help. This is quite important because the authors do claim that this is a more active catalyst, so they need to prove that their method of calculating the sites is indeed reliable!

2. The Arrhenius plots in Figure 3 are not entirely dependable. The conversions are not differential and the plots for the three catalysts are not exactly at identical temperature ranges. Is a difference of 20 kJ/mol really statistically significant considering these issues?

3. There is not much discussion on the role of CO₂ and the support. Clearly CO₂ conversion is higher than the conversion of propane to propylene. This might indicate that (1) CO₂ is reacting with coke produced or (2) dissociating on CeO₂ in addition to potentially reacting with the hydrogen produced from dehydrogenation through a reverse water gas shift reaction. This has not been carefully analyzed. What happens when CO₂ is not co-fed? Is deactivation faster? If yes, can the extra coke that is formed without CO₂ cofeed (with respect to the case with CO₂ cofeed) be quantified and related to CO₂ conversion (over and above propane conversion)? While deactivation is slower on the HEI, it may just mean that the coke that is formed is getting consumed by CO₂.

4. If CeO₂ is supplying oxygen to oxidize the coke that gets formed and is therefore an important component, this needs to be examined more carefully. How is the oxygen on CeO₂ getting replenished? Via CO₂ dissociation? Would deactivation be lower for a CeO₂-supported catalyst in general in the absence of CO₂ as well?

5. While the computational approach has been described in detail (although this reviewer could not find any d-band center or electronic structure information in the results/Sl even though the authors describe the method), the DFT calculations and their purpose have not been explained well. For instance, how was the model for HEI chosen – is it just one random configuration (as the language seems to indicate) or was a rigorous combinatorial analysis done to identify the most stable bulk structure and then the surface facets were constructed from that? Why only evaluate the binding strength of propylene vs its dehydrogenation? This reviewer understands that this represents selectivity to propylene vs over-dehydrogenation (which eventually leads to coke) but authors also claim this catalyst is more active (so C-H activation of propane should be equally important). Further, why compute CO₂ dissociation on a clean surface? Is CO₂ dissociation to CO + O the rate determining step for CO₂ conversion? Is there any independent evidence for this? Can it not occur on CeO₂?

5. The authors do say that the catalyst is regenerable but showing regeneration over multiple cycles (3-5) is probably necessary to really prove the stability.

Reviewer #1:

This manuscript studies CO₂-ODP catalyzed by PtCoNiInGaSn/CeO₂ with a high-entropy intermetallic structure. Very impressively, highly dispersed PtCoNiInGaSn over CeO₂ is obtained via the simple aqueous impregnation by using chloride or nitrate salts as the metal precursors. Moreover, a high space-time yield of propene around 40 mL C₃H₆·min⁻¹·g_{cat}⁻¹ (Figs. 2i and S12) and a relatively high stability are obtained for CO₂-ODP over the PtCoNiInGaSn/CeO₂ catalyst. These merits make the developed catalyst very promising for CO₂-ODP, which is worthy to be published. However, some key issues must be clarified before any publications as commented as follows.

Response: Thank you for the positive evaluation on our manuscript and many important suggestions. We modified the MS and SI based on the comments and made point-by-point response as follows.

1. HEI properties.

1.1 Following the authors' definition stated in the manuscript, "HEIs are multi-metallic alloys with five or more elements and specific crystal structures originating from the parent binary intermetallics (Scheme 1)", what is the crystal structure of PtCoNiInGaSn over the studied oxide supports? What is the crystal structure of HEI over Al₂O₃ or SiO₂ with the intense XRD peaks at 41.8 and 44.1°?

Response: The assignment of these peaks has been written in the original manuscript, (p6, line 108) "When Al₂O₃ or SiO₂ was used as a catalyst support for HEI, two intense peaks appeared at 41.8° and 44.1°, which corresponded to the 102 and 110 diffractions of intermetallic PtSn (Fig. 1b). The shifts in diffraction angles from pristine PtSn can be attributed to changes in the lattice constant caused by site-specific multi-metallization."

1.2 The characterization results confirm that Ga over HEI/CeO₂ is mostly oxidized states rather than the metallic state. In this case, can PtCoNiInGaSn still be HEI? Cannot PtCoNiInSn be formed as HEI? If it can, what will happen for both the crystal structure and the catalytic performance?

Response: Partly yes. As shown in XANES, Ga was mostly oxidized, but not fully. Actually, our EXAFS analysis revealed a small contribution of the Pt–Ga and Ga–Ni/Co metallic bonds. Therefore, we have considered that small amount of metallic Ga remains in the HEI nanoparticles and a large part of Ga are present as oxide out of the HEI nanoparticles, which has been written in the original version. We also performed a control experiment using PtCoNiSn_{1.5}In_{1.5}/CeO₂ and compared with PtCoNiSnInGa/CeO₂ as shown below. The Ga-less catalyst showed lower catalytic performance than the senary HEI, demonstrating the necessity of Ga in the HEI for the best performance. This figure was provided in SI with the corresponding comment in MS as follows: (p9, line 184) "We also investigated the effect of multi-metallization at the Sn site with PtCoNiSn/CeO₂ (Pt: Ni: Co: Sn = 1:1:1:3) and PtCoNiSnIn/CeO₂ (Pt: Ni: Co: Sn = 1:1:1:1.5:1.5), which resulted in lower catalyst stability than HEI/CeO₂, respectively (Supplementary Figs. 10 and 11)."

Supplementary Fig. 11. Catalytic performance of PtCoNiInSn/CeO₂ (Pt:Co:Ni:Sn:In = 1:1:1:1.5:1.5) and HEI/CeO₂ in CO₂-ODP. **a** C₃H₈ conversion, **b** CO₂ conversion, **c** net C₃H₆ selectivity considering CO formed from C_xH_y, and **d** C₃H₆ selectivity in C_xH_y.

1.3 As shown in Fig. 1b, two intense peaks for HEI over Al₂O₃ or SiO₂ are observed at 41.8 and 44.1° while no XRD diffractions are detected for HEI over CeO₂. The authors explain this as that the mean particle size of the metallic species over HEI/CeO₂ is significantly too small to be detectable by XRD. This explanation contradicts the results of the CO chemisorption. According to the dispersion shown in Fig. 2e and Table S4, the particle sizes must be similar for the metals over Pt-Co-In/CeO₂, HEI/CeO₂, and HEI/Al₂O₃ with almost the same metal dispersion of ~30%. Due to the much smaller metal dispersion of 19.7% for HEI/SiO₂, a much larger particle size than that of HEI/CeO₂ or HEI/Al₂O₃ is reasonably expected. The particle size based on CO chemisorption must be calculated and discussed reasonably.

Response: Thank you for the important suggestion. Yes, the reviewer is right. We did not correctly interpret this phenomenon. Generally speaking, it seems that diffraction peaks of metal nanoparticles on CeO₂ tend to be observed much smaller or invisible than those on other oxide supports. The same was also observed in our previous study of Pt-Co-In/CeO₂. Therefore, we now interpret that this is due to the large atomic weight of Ce, which much more strongly scatters X-ray than Si or Al. To say, the diffraction from nanoparticles on CeO₂ becomes much weaker than those on SiO₂ or Al₂O₃, thus resulting in very low or undetectable peaks. The corresponding sentences were revised as follows: (p6, line112) “One possible interpretation is that the reflection of X-ray on the nanoparticles was significantly weakened by the strong X-ray scattering by Ce, of which atomic weight and the scattering factor are much larger than Si and Al.”.

2. Experimental

2.1 Product analysis. "The gas phase was analyzed and quantified using a downstream-equipped online thermal conductivity detection gas chromatograph (Shimadzu GC-8A, 300 column: Gaskuropack 54). C₃H₆, C₂H₄, C₂H₆, CH₄, CO, CO₂, H₂O, and H₂ were detected as reaction products in the outlet gas of all catalysts." Are all of these listed species analyzed by GC with the TCD detector? Can these species be well separated with the GC column? What are the parameters of the GC column? How is the unconverted C₃H₈ analyzed?

Response: We are sorry for the misdescription including H₂ and H₂O, which were not analyzed in this study. But other gases were detected using a single column. The description in experimental section was revised as follows: "The gas phase was analyzed and quantified using a downstream-equipped online thermal conductivity detection gas chromatograph (Shimadzu GC-8A, column: Gaskuropack 54, 80/100 SUS 2m×3mm I.D.). C₃H₈, C₃H₆, C₂H₄, C₂H₆, CH₄, CO, and CO₂ could be separated under the following condition (see Supplementary Fig. 28 for a GC chart): carrier gas; He 30mL min⁻¹, column temperature; 50°C (constant)."

Supplementary Fig. 28. An example of the GC chart for the outlet gas in CO₂-ODP.

2.2. Equations. There are 19 equations for defining conversions, selectivity, TOF, etc. However, many of the results such as the carbon balance, the oxygen balance, and TOF are not provided and discussed. What is the purpose for these definitions? How are the outlet flow rates of different species determined? measured or calculated?

Response: Thank you for the careful check and we are sorry again for the misdescription including unnecessary equation, which were copied from previous publication. We have already deleted it and removed from the experiment part. The outlet gas was measured and the flow rate of each component was calculated based on the GC peak area with calibration to molar ratio. This explanation was also added in SI, the experimental section.

2.3 TPSR. What is m/z used for monitoring CO₂? Is CO simultaneously monitored with the Mass Spectrometer? Is there no CO produced during He-TPSR?

Response: As noted in the caption of Fig. 3, we used m/z of 44 to monitoring the evolution of CO₂. CO was also simultaneously evolved monitoring m/z of 28 as follows. Consistently, CO evolution on HEI occurred at lower temperature. This figure was added in SI.

Supplementary Fig. 15. CO evolution (m/z =28) in He-TPSR on the coked Pt–Co–In/CeO₂ and HEI/CeO₂ catalysts (used in CO₂-ODP for 50 h).

3. Catalytic results.

3.1 The asymmetric conversions of C₃H₈ and CO₂. According to the stoichiometry, the conversions of C₃H₈ and CO₂ for the CO₂-ODP reaction should be equal when the feed molar ratio of C₃H₈ to CO₂ of 1 is applied during the experiments. As shown in Figs. 2a, 2c, 2f, and S8-10, the CO₂ conversion is significantly higher than that of C₃H₈ over the optimal HEI/CeO₂ catalyst, especially in the case of Fig. S10, indicating the severe side reactions. However, the C₃H₆ selectivity of larger than 90% is still obtained (Figs. 2a, 2c, 2f, and S8-10). Why? What are the experimental errors? What is the carbon balance of the experiments? These contradictory results must be carefully checked together with the experimental details missed in the manuscript, which have already commented.

Response: This is because dry reforming of C₃H₈ occurred as a side reaction ($C_3H_8 + 3CO_2 \rightarrow 6CO + 4H_2$), which converted three times as much CO₂ as propane. Therefore, a 10% loss in net C₃H₆ selectivity (90% in Fig. S8c) corresponds to a 30% increase in CO₂ conversion. For example, for HEI/CeO₂, C₃H₈ and CO₂ conversion were 26% and 34% at 20 h (Fig. S8a, b), where the latter is about 1.3 times higher than the former, thus being consistent. The carbon balance was shown in Supplementary Fig. 8 as shown below. We considered three carbon balances; total C balance, C balance in CO_x and C_xH_y. C balance in CO_x is higher than that in C_xH_y because CO is also formed from C_xH_y. Although this trend is prominent at the initial stage of reaction, this becomes minor with reaction time. However, total C balance is always close to unity, indicating that the material balance is well satisfied. The following comments were added: (p7, line156) "(carbon balance was also close to unity, Supplementary Fig. 8)".

Supplementary Fig. 8. Carbon balances in CO₂-ODP over the HEI/CeO₂ catalyst. C balance in CO_x was higher than unity while that in C_xH_y was lower, indicating that dry reforming of propane occurred as a side reaction. Although this trend was prominent at the initial stage of the reaction (<5 h), it became minor with time. However, total carbon balance was always close to unity, which is consistent with the very low coke selectivity.

3.2 The caption for b and c of Fig. 3 is interchanged. Moreover, the description of temperatures for CO₂-TPSR over the coked Pt–Co–In/CeO₂ and HEI/CeO₂ catalysts is not matched with Fig. 3c.

Response: We appreciate the careful check on the manuscript. The suggested mistakes were appropriately corrected.

3.3 The specific activity in Fig. 2i. In fact, it is the space-time yield of propene. I cannot find how around 40 mL_{C₃H₆}·min⁻¹·g_{cat}⁻¹ (Figs. 2i and S12) are obtained. The results must be carefully checked and confirmed. What is the time on stream for the calculation? Which experimental results are used for the calculation?

Response: According to the reviewer's comment, we added the definition of the specific activity in SI as follows (p16, line 377). For the calculation, the initial propylene yield was used.

$$\text{specific activity (mL}_{\text{C}_3\text{H}_6} \text{ min}^{-1} \text{g}_{\text{AM}}^{-1}) = \frac{F_{\text{C}_3\text{H}_8}^{\text{in}} \times Y_{\text{C}_3\text{H}_6}}{W_{\text{AM}}} \quad (18)$$

where, $F_{\text{C}_3\text{H}_8}^{\text{in}}$, $Y_{\text{C}_3\text{H}_6}$, and W_{AM} , and indicate propane inlet flow rate, the initial net propylene yield, and the weight of the active metal (AM) component, respectively. Please note that, in revision, we changed the denominator of this scale from W_{cat} to W_{AM} because we noticed that W_{AM} was more suitable to fairly evaluate various systems with different loading amount or fraction of the active metal species. The specific activity of our catalyst was recalculated as 1414 mL_{C₃H₆} min⁻¹g_{AM}⁻¹, which can be obtained using $F_{\text{C}_3\text{H}_8}^{\text{in}} = 5 \text{ mL min}^{-1}$, $Y_{\text{C}_3\text{H}_6} = 0.283$, and $W_{\text{AM}} = 0.1 \text{ g} \times 0.01$ (catalyst amount \times loading amount of Pt). Fig. 2i was updated including the newly added points of recent publication (also addressing the comment 5) as shown below, showing that our catalyst still at the best.

3.4 “Thus, the HEI phase is capable of inhibiting coke formation while also promoting coke combustion,-”. Following the conclusion, the promoted coke combustion will decrease the C₃H₆ selectivity. Why is it still high over the HEI/CeO₂ catalyst?

Response: “Promoting coke combustion” means just the high ability of coke combustion (not the amount). Here, inhibiting coke formation comes first, which increases selectivity and decreases coke amount. Moreover, the smaller amount of coke is combusted more smoothly. These simply indicate a double promotion effect.

4. Title. What is the meaning of auto-regenerable catalyst? How is it reflected from the results?

Response: This meant in-situ coke combustion by the cooperation of CeO₂, CO₂, and HEI. However, as the reviewer implied, this may be ambiguous, so we deleted this word from the title: “High-entropy intermetallics on ceria as a highly efficient catalyst for the oxidative dehydrogenation of propane using CO₂”.

5. References. In fact, quantitative works over different catalysts have been reported for CO₂-ODP, especially in recent years. However, it is strange that the authors cite only some purposefully selected works before 2010 both in the references and the comparison in Tables S5 and S6. On the contrary, the authors’ many recent works including PDH are cited without any further comparison/discussion in the manuscript. Thus, it is unfair that the important works in recent 5 or 10 years and the very recent review work (ACS Catal., 2021, 11: 2182 – 2234) are not cited, and compared.

Response: Thank you for the kind suggestions. According to the comment, we newly added the results of very recent papers with various temperatures to Fig 2 in MS and some supplementary data in SI (Tables 5, 6, and Fig. 14). Still, our catalyst shows the best performance in various viewpoints.

Reviewer #2:

Xing et al. discuss the performance of a multimetallic high entropy intermetallic (HEI) catalyst on ceria support (and variants of this) for CO₂-oxidative dehydrogenation of propane. The catalyst seems active, selective, and stable. The authors compare the proposed HEI catalyst with various alloys and support combinations to show that the reported catalyst is optimal in terms of selectively producing propylene and generating very little coke. While the work clearly shows an interesting class of materials, the experimental and computational analysis of the catalyst in terms of the active sites and reaction mechanism is incomplete or not well discussed. Specific comments are given below

Response: Thank you for the positive evaluation and insightful comments on our manuscript. The suggested points are indeed important; therefore, we fully addressed them. We believe it greatly improved the scientific reliability of this study.

1. How is the turnover frequency calculated? This reviewer guesses it is possibly using the reported CO chemisorption data. But can chemisorption effectively count sites for the HEI catalyst? It may only bind to certain Pt sites or ensembles (or it may bind to non-Pt ensembles too where the reaction may not take place). A DFT calculation of where CO can bind would help. This is quite important because the authors do claim that this is a more active catalyst, so they need to prove that their method of calculating the sites is indeed reliable!

Response: Yes, the TOF was calculated based on the amount of CO chemisorbed on the active metals. In this study, not only Pt but also Co/Ni act as active metals and they are known to strongly adsorb CO. Conversely, typical metals are known to be inert for CO chemisorption because of the lack of DOS near the Fermi level. According to the comment, we newly added the result of DFT calculation for the CO adsorption energy on each metal's atop site in HEI (Supplementary Fig. 29). As we expected, the adsorption strength of CO was as follows: Ni > Co > Pt >>> In, Sn ~ 0. This result support our assumption for TOF calculation.

(p. 17, line 382) “We confirmed by DFT calculation that CO can strongly adsorb on Pt, Co, and Ni sites (Supplementary Fig. 29).”

Supplementary Fig. 29. E_{ad} (eV) of CO on some atop sites of HEI(001):B. Yellow dotted circles indicate the adsorption sites of CO. Upper insets show the optimized structure of CO on Co and Pt atop sites. Sn and In showed very small negative values corresponding to only physisorption. The largely negative values for Pt, Ni, and Co indicate that they are active for CO pulse chemisorption and should be included in the estimation of metal dispersion and TOF.

2. The Arrhenius plots in Fig. 3 are not entirely dependable. The conversions are not differential and the plots for the three catalysts are not exactly at identical temperature ranges. Is a difference of 20 kJ/mol really statistically significant considering these issues?

Response: Please note that all the kinetic experiments were conducted under differential conditions (conversions below 10%) using lower catalyst amount (please see the experimental details). The temperature range was modified to be within the same ranges, but the order of apparent activation energies was not changed and was still consistent with the results of the DFT calculation. Fig. 3 was updated as follows.

Fig. 3: Mechanistic study. Arrhenius-type plots for **a** C_3H_8 and **b** CO_2 conversion rates obtained in CO_2 -ODP on HEI/CeO₂. **c** He- and **d** CO_2 -TPSR on the coked Pt–Co–In/CeO₂ and HEI/CeO₂ catalysts (used in CO_2 -ODP for 50 h). CO and CO_2 evolutions were quantified by the mass intensity of $m/z = 28$ and 44, respectively.

3. There is not much discussion on the role of CO₂ and the support. Clearly CO₂ conversion is higher than the conversion of propane to propylene. This might indicate that (1) CO₂ is reacting with coke produced or (2) dissociating on CeO₂ in addition to potentially reacting with the hydrogen produced from dehydrogenation through a reverse water gas shift reaction. This has not been carefully analyzed. What happens when CO₂ is not co-fed? Is deactivation faster? If yes, can the extra coke that is formed without CO₂ cofeed (with respect to the case with CO₂ cofeed) be quantified and related to CO₂ conversion (over and above propane conversion)? While deactivation is slower on the HEI, it may just mean that the coke that is formed is getting consumed by CO₂.

Response: Thank you for the important suggestion. The higher CO₂ conversion is due to the occurrence of dry reforming of propane ($C_3H_8 + 3CO_2 \rightarrow 6CO + 4H_2$), which converted three times as much CO₂ as propane. Therefore, a 10% loss in net C₃H₆ selectivity (90% in Fig. S8c) corresponds to a 30% increase in CO₂ conversion. For example, for HEI/CeO₂, C₃H₈ and CO₂ conversion were 26% and 34% at 20 h (Fig. S8a, b), where the latter is about 1.3 times higher than the former. Therefore, the material balance is well satisfied without considering coke, indicating that the intrinsic coke selectivity is very low. However, as the reviewer anticipated, the deactivation is actually fast when CO₂ was not co-fed because coke cannot be removed and is accumulated with time. The results of this control experiment were added in SI as shown below:

Fig. 3. Changes in propane conversion in the DDP-regeneration cycles over **e** HEI/CeO₂ and **f** HEI/SiO₂. Reaction conditions: DDP; C₃H₈:He = 5:10 mLmin⁻¹ (2 h), regeneration; CO₂:He = 10:10 mLmin⁻¹ (1 h), then H₂:He = 5:10 mLmin⁻¹ (0.5 h) at 600°C.

Supplementary Fig. 16. Changes in propane conversion in the DDP-regeneration cycles over HEI/Al₂O₃. Reaction conditions: DDP; C₃H₈:He = 5:10 mLmin⁻¹ (2 h), regeneration; CO₂:He = 5:10 mLmin⁻¹ (1 h), then H₂:He = 5:10 mLmin⁻¹ (0.5 h) at 600°C.

The following comments were added in MS: (p11, line 231) “We further tested the catalytic performance of HEI in DDP as a control experiment without CO₂. As shown in Fig. 3e, rapid deactivation occurred within 0.5 h even for HEI/CeO₂, indicating that simultaneous CO₂ supply is necessary for the continuous coke combustion. This is because the oxygen vacancy of CeO₂ must be refilled for the continuous combustion. In this regard, we previously confirmed by H₂-TPR and CO₂-TPO that the oxygen atoms derived from CO₂ refilled the oxygen vacancy of CeO₂.¹⁴ Then, C₃H₈ conversion was fully recovered when HEI/CeO₂ was regenerated under flowing CO₂ and the subsequent H₂ reduction, demonstrating that coke accumulated on the catalyst can be completely removed. Conversely, the catalytic activity of HEI/SiO₂ (Fig. 3f) and HEI/Al₂O₃ (Supplementary Fig. 16) was not fully recovered after the regeneration process. Therefore, the oxygen releasing ability of CeO₂ is essential for the coke combustion and the high regenerability. HEI/SiO₂ showed slower deactivation rate than HEI/CeO₂, which may be due to much higher specific surface area of SiO₂ than CeO₂.”.

4. If CeO₂ is supplying oxygen to oxidize the coke that gets formed and is therefore an important component, this needs to be examined more carefully. How is the oxygen on CeO₂ getting replenished? Via CO₂ dissociation? Would deactivation be lower for a CeO₂-supported catalyst in general in the absence of CO₂ as well?

Response: The deactivation rate does not correctly reflect the oxidation ability of CeO₂ because the specific surface area of the oxide support also contributes to the coke tolerance (the wider the area is, the more the coke can be accumulated and the longer the life is). As seen in the figure above, the important thing is that coke on CeO₂ can be fully removed by regeneration (CO₂ treatment), whereas coke on SiO₂ and Al₂O₃ cannot, which strongly indicates that CeO₂ is capable of coke combustion. After the oxygen release from CeO₂, the oxygen vacancy on CeO₂ will be refilled by the oxygen from CO₂ activated on the alloy. This behavior has already been confirmed experimentally in our previous study using an “CO₂ titration technique” on Pt/CeO₂ (ref. 14). As shown in the right figure, oxygen vacancy was first made by H₂-TPR. Then, CO₂-TPO was done to observe CO evolution, which confirmed that oxygen was left on the catalyst. Finally, the second H₂-TPR showed again the reduction of CeO₂, demonstrating that oxygen from CO₂ surely filled the vacancy. The response and newly added sentences for the comment 3 can also be the answer to the comment 4.

5. While the computational approach has been described in detail (although this reviewer could not find any d-band center or electronic structure information in the results/SI even though the authors describe the method), the DFT calculations and their purpose have not been explained well. For instance, how was the

model for HEI chosen – is it just one random configuration (as the language seems to indicate) or was a rigorous combinatorial analysis done to identify the most stable bulk structure and then the surface facets were constructed from that? Why only evaluate the binding strength of propylene vs its dehydrogenation? This reviewer understands that this represents selectivity to propylene vs over-dehydrogenation (which eventually leads to coke) but authors also claim this catalyst is more active (so C-H activation of propane should be equally important). Further, why compute CO₂ dissociation on a clean surface? Is CO₂ dissociation to CO + O the rate determining step for CO₂ conversion? Is there any independent evidence for this? Can it not occur on CeO₂?

Response: Thank you for the important suggestions. First, we generated a bulk (2x2x2) model with random atomic distribution with the following restrictions: (1) Pt/Co/Ni occupy the Pt sites while Sn/In do the Sn sites, (2) Pt atoms do not neighbor each other on all the (110) layers, and (3) the number of each metal in each (110) layer is fixed to 2 or 3 (Pt/Ni/Co) and 4 (Sn/In). The distribution of each atom was determined using random numbers. Then, we generated ten different bulk (2x2x2) unit cells, of which energy difference was < 0.5 eV, indicating that the distribution does not strongly influence the stability of bulk structure. Among them, we chose the most stable structure as an energetically likely structure, shown in Fig. 4. The following explanation was added in the computational details section: (p19, line438) “As the parent structure of HEI, a PtSn-(2x2x2) supercell was constructed so that the (110) plane of PtSn corresponded to the (004) plane of the supercell. The distribution of each constituent metal in HEI was determined with the following restrictions: (1) Pt/Co/Ni occupy the Pt sites while Sn/In do the Sn sites (Ga was not included in this model for simplicity because of the low concentration mentioned above), (2) Pt atoms do not neighbor each other on all the (110) layers, and (3) the number of each metal in each (110) layer is fixed to 2 or 3 (Pt/Ni/Co) and 4 (Sn/In). The distribution of each atom was determined using random numbers. Then, we generated ten different bulk (2x2x2) unit cells, of which energy difference was < 0.5 eV, indicating that the distribution does not strongly influence the stability of bulk structure. Among them, we chose the most stable structure as an energetically likely model as shown in Supplementary Table 7 and Supplementary Fig. 17.”.

We also newly calculated the first and second C–H activation steps for activity trends. For CO₂ conversion, chemisorption step to form the bended CO₂ was also considered. The order of E_A was PtSn > PtCoIn > HEI, which was consistent with that of the experimental apparent E_A . This also indicates that HEI is not only selective but also active for PDH. CO₂ chemisorption step was much easier than CO₂ activation for all the catalysts; therefore, CO₂ activation is likely to be rate-determining in CO₂ reduction. In fact, in the previous study, we have revealed by microkinetic modeling that CO₂ activation is the rate-determining step of the overall CO₂-ODP on PtCoIn (ref. 14). The following data and comments were added in the revised MS and SI: (p12, line273) “The E_A of propane to propylene and CO₂ reduction was also calculated on each of the three surfaces. The summary of E_A and the corresponding energy diagrams are shown in Figs. 4d and 4e, respectively (see Supplementary Figs. 18~27 for each structure). HEI showed lower E_A of propane dehydrogenation than PtSn and Pt–Co–In, indicating that the desired C–H activation was enhanced by multi-metallization while the undesired was inhibited. The order of E_A (HEI < Pt–Co–In < PtSn) was consistent with that of the experimental apparent E_A shown in Fig. 3a.

For CO₂ reduction, CO₂ is first chemisorbed on the metallic surface to form a bidentate *sp*²-like conformation, then C–O cleavage occurs to generate CO and O (Fig. 4e, see Supplementary Figs. 15, 21, 22, 23, and 24 for details). Although the chemisorption step showed moderate energy barriers (~ 0.7 eV), they were much lower than those of the CO₂ activation step (1.0~1.3 eV), indicating that CO₂ activation is the rate-determining step (RDS) in CO₂ reduction. In this regard, we recently revealed by microkinetic modeling that CO₂ activation was the RDS in CO₂-ODP over Pt–Co–In.¹⁴”.

Finally, we also calculated the d band structures and provided the results and discussion on the effect of electronic state on the catalysis. As shown in Fig. 4, intermetallic PtSn has low d-DOS near the Fermi level due to alloying with Sn, which may have resulted in the high E_A . Conversely, HEI has an intense peak near the Fermi level owing to the Ni and Co 3d states. Therefore, the d band was significantly upshifted near the Fermi level by doping Ni and Co. The enhanced activity in PDH and CO₂ reduction can be explained by the d band modification by Ni and Co. The following comments were added in MS: (p13, line287) “We also considered the effect the electronic structure changes upon multi-metallization on the catalytic activity. Fig. 4f shows the density of states projected on d orbitals (d-DOS) of the surface transition metals on HEI(001):B and PtSn(110). PtSn had low d-DOS near the Fermi level due to alloying with Sn, which may have resulted in the high E_A . Conversely, HEI showed an intense peak near the Fermi level owing to the Ni and Co 3d states. Therefore, the d band was significantly upshifted near the Fermi level by doping Ni and Co. The enhanced activity in propane dehydrogenation and CO₂ reduction can be explained by the modification of the d band by the introduction of Ni and Co. This demonstrates our catalyst design concept mentioned in the introduction paragraph (Scheme 1).”.

Fig. 4: DFT calculations. **a** Model structures of PtSn and the PtSn-based HEI for DFT calculations. **b** An example of the HEI slab model for CO₂-ODP for the over dehydrogenation: C₃H₆ → C₃H₅+H at a Pt₁-HEI site on the (110) surface

(HEI(001):B3). **c** Comparison of EA of the third C–H activation (overdehydrogenation of C_3H_6), $-E_{ad}$ (E_d) of C_3H_6 , and ΔE ($= E_A + E_{ad}$) on the surface of PtSn, Pt–Co–In, and HEI (average). **d** E_A of propane dehydrogenation and CO_2 activation. For propane dehydrogenation, the higher E_A of the two C–H activation steps was shown (the first C–H: PtSn, HEI, the second C–H: Pt–Co–In). **e** Energy diagrams of propane dehydrogenation and CO_2 reduction on the surface of PtSn, Pt–Co–In and HEI. **f** d-DOS of the surface transition metals on HEI(001):B and PtSn(110).

6. The authors do say that the catalyst is regenerable but showing regeneration over multiple cycles (3-5) is probably necessary to really prove the stability.

Response: According to the reviewer's comment, we additionally tested the third cycles with the second regeneration on HEI. We confirmed that C_3H_8 and CO_2 conversions were fully recovered also after the twice reaction-regeneration cycles. Therefore, our catalysts are surely regenerable and tolerant to long-term use.

REVIEWER COMMENTS

Reviewer #1 (Remarks to the Author):

Just as I commented on the original manuscript, the relative higher performance of the catalysts makes the manuscript worthy to be accepted although the catalytic understanding is much few. However, many careless description, interpretation, and errors are here and there throughout the manuscript, which impede the understanding and consume much time. It cannot be denied that the quality of the revised manuscript is improved. Unfortunately, the same thing is still there throughout the revised manuscript, e.g., The unit of catalyst life in Fig. 2i should be h; The "Time" in Table S6 for calculating CO₂ util. eff. and FC_{3H6}/W_{cat} is the same with that in Table S5; References 15, 18, and 28 missing page number. Thus, the manuscript must be carefully checked before any publications.

Moreover, the issues for replying my specific comments/questions are given as follows.

1.1 What is the crystal structure of HEI? Cubic?

1.3 There are great numbers of articles dealing with noble metals (alloy) over CeO₂ in the open literature. Can the authors read the relevant works, and make the explanation with reference(s)?

2.2. The authors did not fully respond to my comments, e.g., two different expressions of C_{3H6} selectivity and yield are defined for comparison with the literature data, but no one can understand the later discussion, i.e., how are the expressions determined? What are the data used?

More importantly, careless errors still occur in the equation 11, i.e., the formula for calculating the total carbon balance is wrong.

Besides, if the amount of CO from C_{3H8} is added to equation 9 rather than equation 10, is not it much clearer for differentiating the carbon loss from the feed gas of respective C_{3H8} or CO₂?

What is the necessity or importance by defining the CO₂ utilization efficiency?

2.3 In fact, besides m/z of 44, m/z of 28 is also the characteristic peak of CO₂, which is the same to that of CO. How do the authors differentiate the contribution of CO₂ in m/z of 28 to that of CO?

3.1 In fact, the difference between the conversions of C_{3H8} and CO₂ is not significant at the later stage of the reaction, e.g., TOS of 20 h as the author explained. On the contrary, it is very significant at the early stage of the reaction, e.g., TOS of less than 10 h, Why?

3.3 The authors did not directly reply to my comments. On the contrary, they made new calculations with an alternative denominator in the equation, i.e., the weight of the active metal rather than that of the catalyst. Following the calculated example, 1414 mL_{C_{3H6}}·min⁻¹·g_{AM}⁻¹ is equivalent to 14.14 mL_{C_{3H6}}·min⁻¹·g_{cat}⁻¹, which is still very far below that of ~40 mL_{C_{3H6}}·min⁻¹·g_{cat}⁻¹ given in Fig. S14b. What is the time on stream for the calculation? Which experimental results are used for ~40 mL_{C_{3H6}}·min⁻¹·g_{cat}⁻¹ in Fig. S14b?

3.4 I do not agree with the reply on my comments since that there is no any evidence to support the authors' explanation.

Additional issues:

1. Scheme 1c. According to the Scheme, the role of Pt is to activate the C-H bond. However, according to the reaction results in Fig. S9, Pt/CeO₂ is a good catalyst for activating both C-H and C-O bonds in CO₂.

This contradiction should be explained.

2. Figs. 2a and 2b. The corresponding net C₃H₆ selectivity cannot be found in SI. Moreover, it is strange that the C₃H₆ selectivity is decreased at the later stage of the reaction, especially for PtSn and PtCoIn. What happened?

3. Fig. 2g. which C₃H₆ selectivity is used, equations 5 or 6?

4. TPSR in Figs. 3c, 3d, and S15. For He-TPSR of the spent HEI/CeO₂, the release of CO₂ is explained as the coke combustion with the lattice oxygen of CeO₂. However, the starting temperature for releasing CO₂ over HEI is as low as 200°C (Fig. 3c), indicating that the lattice oxygen over CeO₂ in the spent catalyst is very active by following the authors' explanation. If it is so, can coke be deposited over HEI/CeO₂ at the very high temperature of 600°C during the catalytic reaction? Abnormally, the starting temperature for releasing CO over the spent HEI/CeO₂ during He-TPSR (~500°C, Fig. S15) is obviously lower than that during CO₂-TPSR (~600°C, Fig. 3d), indicating the easier coke removal under He atmosphere. These abnormal observations are very possibly caused from impurities such as O₂ during He-TPSR, which is worthy to be checked.

Reviewer #2 (Remarks to the Author):

The authors have carefully addressed the comments raised by this reviewer. DFT modeling of HEI catalysts is a tricky business; the authors have done considerable additional calculations in this revision. The net additions (in response to questions raised by this reviewer and the other) clearly improve the quality of the manuscript. This reviewer believes it is acceptable for publication.

Reviewer #1

Just as I commented on the original manuscript, the relative higher performance of the catalysts makes the manuscript worthy to be accepted although the catalytic understanding is much few. However, many careless description, interpretation, and errors are here and there throughout the manuscript, which impede the understanding and consume much time. It cannot be denied that the quality of the revised manuscript is improved. Unfortunately, the same thing is still there throughout the revised manuscript, e.g., The unit of catalyst life in Fig. 2i should be h; The “Time” in Table S6 for calculating CO₂ util. eff. and $F_{C_3H_6}/W_{cat}$ is the same with that in Table S5; References 15, 18, and 28 missing page number. Thus, the manuscript must be carefully checked before any publications. Moreover, the issues for replying my specific comments/questions are given as follows.

Thank you for your re-evaluation and the careful check on our manuscript. We apologize that there remained careless errors and unfriendliness for sufficient understanding. Besides, we really appreciate the reviewer’s comments including those listed below, which helps to understand the important aspects of this study. The errors suggested above were appropriately corrected and we carefully and deeply checked the manuscript to exclude any inconsistency or unclear point. The time on stream for the initial and final values in Table S6 are the same with those in Table S5 (the initial time was used for calculating the CO₂ utilization efficiency). This information was added in Table S6 footnote. The responses to the issues raised below were provided point by point as follows. We believe that the re-revised version of our manuscript was further improved and is now suitable for publication.

1.1. What is the crystal structure of HEI? Cubic?

No. It is a hexagonal NiAs-type structure with the space group of P6₃/mmc. This information was added in the caption of Scheme 1a.

1.3. There are great numbers of articles dealing with noble metals (alloy) over CeO₂ in the open literature. Can the authors read the relevant works, and make the explanation with reference(s)?

According to reviewer’s advice, the following sentences and references were added in the introduction paragraph: (p3, line 58) “Using CeO₂ or Ce-based oxides as a support of the metallic phase is also a promising way for CO₂-ODP owing to various promotional effects such as Mars-van Kreveren-type CO₂ activation²²⁻²⁴ or coke combustion,^{14,25} and strong metal-support interaction to tune the character of the active phase.^{26,27”}.

2.2. The authors did not fully respond to my comments, e.g., two different expressions of C₃H₆ selectivity and yield are defined for comparison with the literature data, but no one can understand the later discussion, i.e., how are the expressions determined? What are the data used? More importantly, careless errors still occur in the equation 11, i.e., the formula for calculating the total carbon balance is wrong. Besides, if the amount of CO from C₃H₈ is added to equation 9 rather than equation 10, is not it much clearer for differentiating the carbon loss from the feed gas of respective C₃H₈ or CO₂? What is the necessity or importance by defining the CO₂ utilization efficiency?

(1) We used the two definitions of C₃H₆ selectivity (yield) for fair comparison with all the reported data because the definition differs depending on literature. Many papers reported the C₃H₆ selectivity as its fraction only among C1-C3 hydrocarbons. However, as mentioned in other papers including this study, dry reforming of propane to CO also occurs, which decreases the net selectivity of C₃H₆ among C1-C3 plus CO. Therefore, it is necessary to show both selectivities (sel. in HC and net sel.) for fair comparison and the same is true for C₃H₆ yield. For the definition of the net C₃H₆ selectivity, the amount of C₃H₈ converted into CO (1/3 of CO generated from hydrocarbon, which can be estimated by

eq. 4) was added in the denominator (eqs. 5 to 6). Since we had missed to include this information in Figures 2h-i and Tables S5-S6, these items were updated.

Besides, the following comments were also added: (p.7, line 150) “Fig. 2a, 2b, and 2c show the time-course of C₃H₈ conversion, C₃H₆ selectivity in hydrocarbons (see Supplementary Fig. 8 for the net C₃H₆ selectivity, which includes CO formed from hydrocarbons via dry reforming), and CO₂ conversions, respectively. With the exception of Pt/CeO₂ (72% sel.), all catalysts converted C₃H₈ by approximately 30% with a high C₃H₆ selectivity (90%–94%). The net C₃H₆ selectivity was comparable to that in hydrocarbons for PtSn and HEI, whereas it was lower for Pt and Pt–Co–In, indicating that dry reforming of propane was inhibited on the PtSn-based structure”, (p.10, line 214) “Here, two selectivity/yield descriptions (filled: in hydrocarbons, open: including CO via dry reforming) are shown for better comparison with those in literature because the description differs depending on literature.”.

- (2) Please note that eq 11 is correct. First, the carbon balances in HC and CO_x are based on the single reactant molecules of C₃H₈ and CO₂, respectively; therefore, there should be no weighting of carbon number. This is why the denominator is described as $F_{C_3H_8}^{in} + F_{CO_2}^{in}$. However, we have to distinguish the origin of CO, whether it is from C₃H₈ or CO₂. Therefore, CO from C₃H₈ should be regarded as one of the fragmented products from C₃H₈ as for other HC, thus being divided by 3, whereas CO from CO₂ should not. Indeed, the values obtained by eq 11 are always very close to unity as shown in Fig. S9.

$$\frac{F_{C_3H_8}^{out} + F_{C_3H_6}^{out} + \frac{2}{3}F_{C_2H_6}^{out} + \frac{2}{3}F_{C_2H_4}^{out} + \frac{1}{3}F_{CH_4}^{out} + F_{CO_2}^{out} + \frac{1}{3}F_{CO}^{C_xH_y} + F_{CO}^{CO_2}}{F_{C_3H_8}^{in} + F_{CO_2}^{in}} \times 100 \quad (11)$$

In this context, using the term of “carbon balance” may lead misunderstanding. Therefore, we changed the word to “material balance”. Adding the amount of CO from C₃H₈ to eq 9 is indeed simpler and is also valid to consider the total material balance in HC. However, considering the total material balance in all the carbon-containing molecules is more important and physically meaningful because we are focusing on the sum of the material balances in HC and CO_x.

- (3) CO₂ utilization efficiency was used as the scale of how much CO₂ is converted (X_{CO_2}) and how close to the 1:1 stoichiometry the consumption of CO₂ and the formation of C₃H₆ (the latter term: maximum (unity) at the 1:1 stoichiometry). The CO₂ utilization efficiency becomes high when CO₂ conversion is high and when the stoichiometry of CO₂ conversion and C₃H₆ formation is close to unity, i.e., CO₂ is solely used for CO₂-ODP. Even if CO₂ conversion is so high, it is less meaning if C₃H₆ formation rate is low. Conversely, excess use of CO₂ typically results in low CO₂ conversion, which also gives low CO₂ utilization efficiency even if C₃H₆ yield is high. The following sentences were added: (p.17, line 395) “The CO₂ utilization efficiency was used as the scale of how much CO₂ is converted (X_{CO_2}) and

how close to the 1:1 stoichiometry the consumption of CO₂ and the formation of C₃H₆ (the latter term: maximum (unity) at the 1:1 stoichiometry) are. The CO₂ utilization efficiency becomes high when CO₂ conversion is high and when the stoichiometry of CO₂ conversion and C₃H₆ formation is close to unity, i.e., CO₂ is solely used for CO₂-ODP. Conversely, it becomes low when CO₂ conversion is much higher than C₃H₆ yield or when excess CO₂ is used.”

2.3. In fact, besides m/z of 44, m/z of 28 is also the characteristic peak of CO₂, which is the same to that of CO. How do the authors differentiate the contribution of CO₂ in m/z of 28 to that of CO?

In our ionization condition, the signal intensity of m/z =28 is only about 6% of that of m/z = 44 when CO₂ alone is flowed without catalyst as shown below. However, for He-TPSR, the signal intensity of m/z =28 is comparable to or higher than that of m/z = 44, indicating that the contribution of CO₂ to the signal of m/z = 28 is negligible. For CO₂-TPSR, the peak feature in the signal of m/z = 28 was not observed in that of m/z = 44, which purely indicates the evolution of CO. This information was added in the method paragraph (p.18, line 431).

Appendix Figure 1 (for review only). Mass signals of the outlet gas: **a** CO₂ flow, **b** He-TPSR (corresponding to Fig. 3c and S15), and **c** CO₂-TPSR on the coked HEI/CeO₂ (corresponding to Fig. 3d).

3.1 In fact, the difference between the conversions of C₃H₈ and CO₂ is not significant at the later stage of the reaction, e.g., TOS of 20 h as the author explained. On the contrary, it is very significant at the early stage of the reaction, e.g., TOS of less than 10 h, Why?

This is also due to the occurrence of dry reforming of propane as we have explained. Please note that the net C₃H₆ selectivity at the initial stage of the reaction is not as high as that after 10 h (75% → 90%, Fig. S10c). The 25% loss of selectivity corresponds to a 75% increase in CO₂ conversion, which is consistent with the initial conversion ratio of 1.71 (CO₂: 53%, C₃H₈: 31%).

3.3. The authors did not directly reply to my comments. On the contrary, they made new calculations with an alternative denominator in the equation, i.e., the weight of the active metal rather than that of the catalyst. Following the calculated example, 1414 mL_{C₃H₆}·min⁻¹·g_{AM}⁻¹ is equivalent to 14.14 mL_{C₃H₆}·min⁻¹·g_{cat}⁻¹, which is still very far below that of ~40 mL_{C₃H₆}·min⁻¹·g_{cat}⁻¹ given in Fig. S14b. What is the time on stream for the calculation? Which experimental results are used for ~40 mL_{C₃H₆}·min⁻¹·g_{cat}⁻¹ in Fig. S14b?

The reviewer’s calculation is correct, and we noticed that we had a careless mistake in that previous calculation (the factor of 3 to compare with Pt–Co–In having 3wt% Pt was carelessly remained in the calculation for comparison with other reported systems). We sincerely apologize for that mistake. Please note that the value in the revised version is surely correct and superior to the reported systems, which has also been shown with the corresponding equations in the previous response letter. To follow this line, Fig. S14 (now S16) was also revised to be shown with the unit of

$\text{mL}_{\text{C}_3\text{H}_6} \cdot \text{min}^{-1} \cdot \text{g}_{\text{AM}}^{-1}$. The time on stream for the calculation of C_3H_6 -related values is 8 h, which was added in Tables S5 footnote.

3.4. I do not agree with the reply on my comments since that there is no any evidence to support the authors' explanation.

We have already shown the evidence: the higher C_3H_6 selectivity (Fig. 2b) and ΔE calculated by DFT (Fig. 4c) support the greater ability of HEI to inhibit coke formation, while the lower peak temperatures in He-TPSR and CO_2 -TPO (Fig. 3c) indicate the greater ability of HEI for coke combustion. First of all, since the coke selectivity is in a much lower level (only 0.001%~0.004%) as shown in Table S3, the difference in the coke combustion ability is too small to be reflected in C_3H_6 selectivity, but only seen in the difference of the catalyst stability and accumulated coke amount.

Additional issues:

1. Scheme 1c. According to the Scheme, the role of Pt is to activate the C-H bond. However, according to the reaction results in Fig. S9, Pt/CeO₂ is a good catalyst for activating both C-H and C-O bonds in CO₂. This contradiction should be explained.

Although Pt/CeO₂ is indeed active for CO₂ conversion, the problem is that the CO₂ activation ability was significantly decreased by alloying with typical metals as observed for PtGa. This trend was also seen in our previous work for Pt-Co-In/CeO₂ (ref. 14). Therefore, the element "Pt" in Scheme 1c indicates atomic Pt (in intermetallic compounds) rather than nanoparticulate Pt. To clarify this point, the following comment was added: (p.4, line 81) "However, alloying with typical metals is known to significantly decrease the CO₂ activation ability of Pt. Therefore, PtSn's Pt site was then partially substituted with Ni and Co to incorporate metals more capable of CO₂ activation. This has another merit to further dilute Pt-Pt sites for higher selectivity."

2. Figs. 2a and 2b. The corresponding net C_3H_6 selectivity cannot be found in SI. Moreover, it is strange that the C_3H_6 selectivity is decreased at the later stage of the reaction, especially for PtSn and PtCoIn. What happened?

The corresponding data was added in SI as Fig. S8.

Supplementary Fig. 8. Net C_3H_6 selectivity considering CO formed from C_xH_y via dry reforming obtained in CO_2 -ODP over monometallic Pt/CeO₂, PtCo/CeO₂, PtNi/CeO₂, PtIn/CeO₂, PtGa/CeO₂, and HEI/CeO₂. Pt/CeO₂ showed an increase in the net C_3H_6 selectivity, even though C_3H_6 selectivity in HC was constant, indicating that dry reforming was inhibited as the reaction proceeded.

As the reviewer suggested, the decrease in C_3H_6 selectivity is indeed strange. A possible interpretation is the segregation of the alloy phase in the harsh condition to form monometallic Pt because C_3H_6 selectivity came close to the similar level to that of Pt/CeO₂. Actually, we newly performed the regeneration on the spent PtSn/CeO₂ in the CO₂-ODP, which fully recovered C_3H_6 selectivity to the original level. The following comments were added to this regard: (p.8, line 179) “The loss of C_3H_6 selectivity at the later stage of the reaction for PtSn and Pt–Co–In may be due to the segregation of Pt from the alloy phase because the final selectivity was close to that for Pt.”, (p.9, line 207) “When the spent PtSn/CeO₂ catalyst underwent the regeneration process including H₂ reduction, C_3H_6 selectivity was recovered to the original level (Supplementary Fig. 14), indicating that the segregated Pt–SnO_x composite was alloyed again. However, the conversion of C_3H_8 and CO₂ was not recovered, which is consistent with the increase in the size of nanoparticles by sintering.”.

Supplementary Fig. 14. Catalytic performance of PtSn/CeO₂ catalyst in the CO₂-ODP before and after regeneration procedure: **a** C_3H_8 conversion, **b** CO₂ conversion, **c** C_3H_6 selectivity in HC, and **d** net C_3H_6 selectivity.

3. Fig. 2g. which C_3H_6 selectivity is used, equations 5 or 6?

This includes both selectivity expressions. To clarify this, the symbols were distinguished to designate which of them as shown below.

4. TPSR in Figs. 3c, 3d, and S15. For He-TPSR of the spent HEI/CeO₂, the release of CO₂ is explained as the coke combustion with the lattice oxygen of CeO₂. However, the starting temperature for releasing CO₂ over HEI is as low as 200°C (Fig. 3c), indicating that the lattice oxygen over CeO₂ in the spent catalyst is very active by following the authors' explanation. If it is so, can coke be deposited over HEI/CeO₂ at the very high temperature of 600°C during the catalytic reaction? Abnormally, the starting temperature for releasing CO over the spent HEI/CeO₂ during He-TPSR (~500°C, Fig. S15) is obviously lower than that during CO₂-TPSR (~600°C, Fig. 3d), indicating the easier coke removal under He atmosphere. These abnormal observations are very possibly caused from impurities such as O₂ during He-TPSR, which is worthy to be checked.

- (1) First, we checked the involvement of O₂ as an impurity during the TPSR experiments. However, the signal corresponding to O₂ (m/z = 32) was below the detection limit as shown in the Appendix Figure 1 mentioned above, which can exclude the involvement of O₂ in the TPSR trend.
- (2) Then, we reconsider the interpretation of the CO₂ evolution from very low temperature according to the reviewer's suggestion. It is also difficult to understand that CO₂, which should be formed by over-oxidation of CO, is formed at much lower temperature than CO (Appendix Figure 1). Now, we attributed it to the contamination of CO₂ in air on the spent catalyst, i.e., we may just see desorption of contaminated CO₂. Indeed, we observed the desorption of CO₂ below 400°C when the fresh HEI/CeO₂ catalyst was used (Appendix Figure 2 shown below).

Appendix Figure 2 (for review only). Mass signals of the outlet gas in He-TPSR (m/z = 44: left, 28: right).

Therefore, we now consider that the CO₂ evolution does not correctly reflect the coke combustion behavior, hence should not be used as an indicator of coke combustion to avoid misunderstanding. Therefore, CO evolution was used instead (previous Fig. S15 was moved to MS to replace Fig. 3c).

(3) For the difference in the starting temperature of CO release, please note that CO₂ must be activated by metallic catalyst to supply oxygen for coke combustion, whereas the lattice oxygen of CeO₂ can directly react with coke. Therefore, there is no necessity that CO₂-TPSR should show lower temperature of coke combustion than He-TPSR. Having said that, there remains a question that why HEI showed lower coke combustion temperature in He-TPSR than Pt–Co–In. This can be explained by the redox property of CeO₂ being enhanced by multi-metallization. As shown in the H₂-TPR (Fig. S4, the magnification was shown in Fig. S17 together with the He-TPSR), the reduction of CeO₂ was also promoted by the multi-metallization as well as that of the alloy phase (a new reduction peak assignable to the reduction of CeO₂ appeared around at 400–550°C). The starting temperature of CO evolution in the He-TPSR on HEI/CeO₂ roughly agree with this region. The same is also true for Pt–Co–In/CeO₂ (both start from 600°C). Thus, the coke combustion ability of the lattice oxygen of CeO₂ seems to follow the reducibility (oxygen releasing ability) of CeO₂. This means that the multi-metallization to HEI enhances not only the coke combustion ability of CeO₂, but also the CO₂ activation ability for the continuous coke combustion. To clarify this point, the following data and sentences were added. (p.11, line 238) “When the temperature was elevated under flowing He, CO was produced, indicating that coke was combusted by CeO₂ lattice oxygen. (He-TPSR, Fig. 3c). Notably, the combustion temperature for HEI was lower than that of Pt–Co–In, demonstrating that the coke combustion ability of HEI/CeO₂ is superior to that of Pt–Co–In/CeO₂. This may be explained by the enhanced redox property of CeO₂ by multi-metallization (see Supplementary Fig. 17 for details). When TPSR was performed on coked HEI/CeO₂ in the presence of CO₂ (CO₂-TPSR, Fig. 3d), a large amount of CO evolved from 600°C and was completely combusted at 700°C. On the other hand, when coked Pt–Co–In/CeO₂ was used, CO evolution occurred at higher temperatures (from 650°C), which is consistent with the trend in E_A of CO₂ activation. Based on these results, we concluded that the high coke resistance of HEI/CeO₂ originated from the enhancement in the redox property of CeO₂ and the CO₂ activation ability of the alloy phase by multi-metallization.”.

Supplementary Fig. 17. H₂-TPR profiles of unreduced HEI/CeO₂ and Pt–Co–In/CeO₂ (magnification of Supplementary Fig. 4) and He-TPSR on the coked Pt–Co–In/CeO₂ and HEI/CeO₂ (magnification of Fig. 3c). The reduction of CeO₂ was promoted by the multi-metallization as well as the reduction of the alloy phase shown in Supplementary Fig. 4 (a new reduction peak assignable to the reduction of CeO₂ appeared around at 400–550°C). The starting temperature of CO evolution in the He-TPSR on HEI/CeO₂ roughly agreed with this region. A similar trend was also observed for Pt–Co–In/CeO₂ (both started from 600°C). Thus, the coke combustion ability of the lattice oxygen of CeO₂ seems to follow the reducibility of CeO₂.

REVIEWERS' COMMENTS

Reviewer #1 (Remarks to the Author):

This is the second revision of the manuscript, and one cannot deny the improved clarity of the expressions with updated definitions. However, the following is still an issue, i.e., (1) some awkward definitions such as eqs. 3-6, 9-11 and careless errors such as “career gas” in the “catalytic test” part are still there, which impede the understanding of the work; (2) the catalytic explanation responded to my comments is still lacking of consistence and reliability. Since that this is the second revision and the review of the work costed me much time, I do not want to waste my time to point the careless issues again, and give the conclusion right to the Editor.

Reviewer #1

This is the second revision of the manuscript, and one cannot deny the improved clarity of the expressions with updated definitions. However, the following is still an issue, i.e., (1) some awkward definitions such as eqs. 3-6, 9-11 and careless errors such as “career gas” in the “catalytic test” part are still there, which impede the understanding of the work; (2) the catalytic explanation responded to my comments is still lacking of consistence and reliability. Since that this is the second revision and the review of the work costed me much time, I do not want to waste my time to point the careless issues again, and give the conclusion right to the Editor.

Response: Thank you again for your careful check on our manuscript. We corrected the spell miss of “career gas” to “carrier gas”. Although the reviewer claims that the definition of these equations are awkward, the use of them has been thoroughly checked and regarded as valid for the evaluation of CO₂-ODP through the recent peer-review process for our previous publication on CO₂-ODP (Pt–Co–In/CeO₂: ref. 14, *Nat. Catal.*, **2022**, 5, 55-65), where four specialists carefully checked the manuscript and equations and contributed to improve the description of these equations. Therefore, we do not think that these definitions impede the understanding of our work. In the previous revision, we sincerely made the response and modification to fully answer the reviewers questions and suggestions. We apologize that we could not satisfy what the reviewer wondered; however, we have provided additional information on the details in SI as much as possible, which will be helpful for readers' understanding.